

# Dependency of the impacts of geoengineering on the stratospheric sulfur injection strategy part 1: Intercomparison of modal and sectional aerosol module

Anton Laakso[1], Ulrike Niemeier[2], Daniele Visioni[3], Simone Tilmes[4], and Harri Kokkola[1]

[1]Finnish Meteorological Institute, Atmospheric Research Centre of Eastern Finland, Kuopio, 70200, Finland
[2]Max Planck Institute for Meteorology, Bundesstr. 53, 20146 Hamburg, Germany
[3]Sibley School of Mechanical and Aerospace Engineering, Cornell University, Ithaca, NY, USA
[4]National Center for Atmospheric Research, Boulder, CO, USA

**Correspondence:** Anton Laakso (anton.laakso@fmi.fi)

**Abstract.** Injecting sulfur dioxide into the stratosphere with the intent to create an artificial reflective aerosol layer is one of the most studied option for solar radiation management. Previous modelling studies have shown that stratospheric sulfur injections have the potential to compensate the greenhouse gas induced warming at the global scale. However, there is significant diversity in the modelled radiative forcing from stratospheric aerosols depending on the model and on which strategy is used to inject

sulfur into the stratosphere. Until now it has not been clear how the evolution of the aerosols and their resulting radiative forcing depends on the aerosol microphysical scheme used, that is, if aerosols are represented by modal or sectional distribution. Here, we have studied different spatio-temporal injections strategies with different injection magnitudes by using the aerosol-climate model ECHAM-HAMMOZ with two aerosol microphysical modules: the sectional module SALSA and the modal module M7. We found significant differences in model responses depending on the used aerosol microphysical module. In a case where

$SO_2$ was injected continuously in the equatorial stratosphere, simulations with SALSA produced 88%-154% higher all-sky net radiative forcing than simulations with M7 for injection rates from 1 to 100 $Tg(S)yr^{-1}$. These large differences are identified to be caused by two main factors. First, the competition between nucleation and condensation: while in SALSA injected sulfur tends to produce new particles at the expense of gaseous sulfuric acid condensing on pre-existing particles, in M7 most of the gaseous sulfuric acid partitions to particles via condensation at the expense of new particle formation. Thus, the effective

radii of stratospheric aerosols were 10-52% larger in M7 than in SALSA, depending on injection rate and strategy. Second, the treatment of the modal size distribution in M7 limits the growth of the accumulation mode which results in a local minimum in aerosol number size distribution between the accumulation and the coarse modes. This local minimum is in the size range where the scattering of solar radiation is most efficient. We also found that different spatial-temporal injection strategies have a significant impact on the magnitude and zonal distribution of radiative forcing. Based on simulations with various injection

rate using SALSA, the most efficient studied injection strategy produced 33-42% radiative forcing compared to the least efficient strategy while simulations with M7 showed even larger difference of 48-76%. Differences in zonal mean radiative forcing were even larger than that. We also show that a consequent stratospheric heating and its impact on the quasi-biennial oscillation depends both on the injection strategy and the aerosol microphysical model. Overall, these results highlight a crucial





role of aerosol microphysics on the physical properties of stratospheric aerosol which in turn causes significant uncertainties
in estimating climate impacts of stratospheric sulfur injections.

## 1 Introduction

Solar radiation management (SRM) techniques are proposed to complement mitigation efforts to avoid greenhouse gas driven
catastrophic global warming (e.g. (Caldeira et al., 2013)). These techniques might be considered if major reductions in green-
house gas (GHG) emissions are not achieved or the development of efficient carbon dioxide removal techniques are delayed.
Instead of altering increased GHG concentration in the atmosphere, which is suppressing outgoing longwave (LW) radiation,
SRM techniques would aim to reflect more shortwave (SW) radiation from the Earth's atmosphere back to space in order
to mitigate GHG-induced changes in the net radiation flux. Even though SRM could be in theory used to mitigate or even
compensate net radiation flux changes due to GHGs, it changes the structure of the Earth's energy budget. This would lead
to some side effects. For example, offsetting GHG-induced warming by reflecting more radiation decreases the global mean
precipitation ((Tilmes et al., 2013; Laakso et al., 2020)) and can lead to the cooling of the tropics while high latitudes are still
warming, if solar radiation is reduced uniformly (etc percent solar constant reduction) (Kravitz et al., 2021).

One of the most cost-efficient technique to increase the reflectivity of the Earth is continuous stratospheric aerosol interven-
tion (SAI) using sulfur. This technique mimics large volcanic eruptions, where a large amount of sulfur reaches the stratosphere,
subsequently forming aerosols from gaseous sulfur that form a long lasting (1-2 years) reflective blanket temporally cooling
the climate. Thus, this is the one of the few or if not the only SRM technique for which we have observational evidence sup-
porting its efficiency in cooling the climate at the global scale. However, due to the rare occurrence of large volcanic eruptions
which has long lasting climate impacts, there are good modern day observations on volcanic aerosols properties and radiative
effects only for one large volcanic eruption, Pinatubo in 1991. In addition, sulfur from large volcanic eruption is released to a
relatively particle free stratosphere. In the case of geoengineering, stratospheric sulfur injections are instead done continuously
onto existing particle field from the preceding injections. This would affect the size distribution of the stratospheric aerosols
and the following radiative and climate impacts. Thus it is not straightforward to draw conclusions about possible impacts of
SAI based on observations of large volcanic eruptions. A lack of measurements have resulted in a major uncertainty on the
estimations of cooling and other impacts of stratospheric sulfur injections (Plazzotta et al., 2018).

Because of the lack of measurements, most of the climate impact estimations of SAI rely on climate model simulations. There
are several approaches to model SAI. In some studies, the effect of stratospheric sulfur injections are imitated by decreasing
the solar constant, which however is not a good proxy for radiative impacts of stratospheric aerosols (Visioni et al., 2021).
Aerosols absorb a part of the longwave radiation which has an impact on the atmospheric energy budget and on atmospheric
dynamics. In addition, the cooling potential and the spatial distribution of radiative forcing depend on aerosol microphysics and
the transport of the particles. Studies where aerosol microphysics is simulated, have shown that global mean radiative forcing
does not increase linearly with the amount of injected sulfur (Heckendorn et al., 2009; Pierce et al., 2010; Niemeier et al.,
2011). Increasing the magnitude of the injection leads to larger particles in size with a smaller number concentration. Larger





particles work as an efficient condensation sink for gaseous sulfuric acid and a coagulation sink for new particles forming via nucleation. However, Tilmes et al. (2018a) have shown a linear relationship between injection amount and temperature reduction of SAI.

There is a large diversity in the predicted radiative forcing of stratospheric geoengineering between studies where aerosols microphysics is simulated. For example, based on Niemeier and Timmreck (2015), the injection rate of 10 Tg(S)yr$^{-1}$ leads to -(1.79-2.06) Wm$^{-2}$ all-sky radiative forcing at the top of the atmosphere (TOA). Based on Laakso et al. (2020), only 3 Tg(S)yr$^{-1}$ were required to induce a radiative forcing of same magnitude (-2.2 Wm$^{-2}$). Injecting 6 Tg(S) per year in the Laakso et al. (2020) resulted in -3.72 Wm$^{-2}$ total radiative forcing while achieving the same cooling effect required

20 Tg(S)yr$^{-1}$ in simulations of Niemeier and Timmreck (2015). Both studies used different generation of the same general circulation model (GCM) ECHAM but the main difference was how the aerosol microphysical processes were modelled. Simulations by Niemeier and Timmreck (2015) were done with a modal aerosol model (M7) while a sectional model (SALSA) was used in Laakso et al. (2020).

     In addition, Kleinschmitt et al. (2018) studied the dependency of geongineering to the magnitude of SAI using a GCM with

a sectional aerosol scheme (LMDZ-S3A). Their results on net radiative forcing where close to the values in Niemeier and Timmreck (2015), but both LW and SW radiative forcing, which have opposite impact on net radiation (i.e different signs), were individually significantly larger. In Niemeier and Timmreck (2015), the LW forcing efficiency (forcing per injected sulfur) were roughly 0.1 Wm$^{-2}$/(Tg(S)yr$^{-1}$) regardless of injection rate. SW forcing efficiency was -0.35 Wm$^{-2}$/(Tg(S)yr$^{-1}$) with 1 Tg(S)$^{-1}$ sulfur injection rate and decreased gradually to -0.22 Wm$^{-2}$/(Tg(S)yr$^{-1}$) when injection rate were increased to 50

Tg(S)$^{-1}$. In Kleinschmitt et al. (2018), the corresponding change in SW forcing efficiency was -0.5 to -0.3 Wm$^{-2}$/(Tg(S)yr$^{-1}$) while the LW forcing varied between 0.2 and 0.3 Wm$^{-2}$/(Tg(S)yr$^{-1}$). Kravitz et al. (2017) simulated injections with strengths between 1-25 Tg(S)yr$^{-1}$ with CESM1(WACCM) where the aerosol size distribution is represented using three modes. In their study, the SW forcing efficiency varied roughly from -1.0 to -0.7 Wm$^{-2}$/Tg(S)yr$^{-1}$ and the LW forcing from 0.7 to 0.6 Wm$^{-2}$/Tg(S)yr$^{-1}$. However, these fluxes were calculated from fully coupled simulations and thus are not fully comparable to

the direct radiative forcing estimates of the other above-mentioned studies.

     In addition to the fact that the simulated radiative effects depend on which model is used, they also depend on the injection strategy and how injections are varied spatially and temporally. There is a good agreement between studies in that when injection rates are lower than 10 Tg(S)yr$^{-1}$, increasing altitude of the injection increases the lifetime of aerosols and the radiative forcing (Heckendorn et al. (2009); Niemeier et al. (2011); Kleinschmitt et al. (2018); Vattioni et al. (2019); Tilmes

et al. (2018b)), but with with higher injection rates this can be the opposite as shown by Niemeier and Schmidt (2017).

     Most of the studies have simulated the impacts of injections over the Equator where yearly average solar intensity is highest. In addition, due to the Brewer-Dobson circulation it takes longer for aerosol to be transported to high latitudes, where sedimentation rate is larger than in low latitudes. Niemeier et al. (2011) showed that injecting only to one model grid box induced a stronger radiative forcing compared to when injections were performed in a band over Equator. However, based on Vattioni

et al. (2019) injecting to one grid box or band over longitudes did not have a significant impact, while (Mills et al., 2017) found that injections along one longitude result in larges particles than point injections and are therefore less efficient. English et al.





(2012) showed that an injection to a broader band over the Equator would increase the lifetime of sulfur, while Vattioni et al. (2019) did not find a significant impact of broadening injection area. On the other hand, Kleinschmitt et al. (2018) found that broadening the injection band had a negligible impact on the net radiative forcing but individually SW and LW forcing, which

have opposite impact on the radiation, decreased by 20-30 % in the case of 10 Tg(S)yr$^{-1}$ injection rate.

Sulfur can also be injected at the certain time of the year instead of continuous injections. Heckendorn et al. (2009) and Niemeier et al. (2011) studied scenarios where injections were done twice per year. Based on Heckendorn et al. (2009), this strategy of using pulsed injections increased the forcing more than 50% compared to continuous injections. However, based on Niemeier et al. (2011) continuous and pulsed injection scenarios did not exhibit a large difference in their radiative impacts.

Visioni et al. (2019) simulated single-point injections at the different latitudes only in certain season and showed that it can reduce the required sulfur for achieving a certain aerosol optical depth. The injection area can also be varied spatially depending on the season. Laakso et al. (2017) and Kleinschmitt et al. (2018) showed that this led to a slight increase of radiative forcing compared to continuous Equatorial injections, but the zonal distribution of the forcing was concentrated relatively more on midlatitudes and less over the Tropics. The sensitivity of modelled response to different spatio-temporal injection strategies

can also be dependent on to the injection magnitude which has not so far been studied.

Overall, as these studies listed above show, there is a large diversity in radiative forcings of SAI between studies and differences depend on which general circulation model and microphysical module is used, how the injections are varied spatially and temporally, and what is the magnitude of the sulfur injections. Simulating aerosol microphysics is computationally heavy. This has sets limitations for investigating different injection scenarios with different amounts of injected sulfur. However, increased

computing capacity in the last few years enables studying a wide range of different injection strategies in a feasible computation time. There are only few aerosol-climate models which include both modal and sectional approaches for representing aerosol size distribution and calculating aerosol microphysics which would allow studying how the the aerosol microphysics scheme affects the simulated impacts of stratospheric sulfur geoengineering. Here we do comprehensive study on the radiative impacts of stratospheric sulfur injections. We use ECHAM-HAMMOZ aerosol-climate model, which includes both the modal aerosol

module - M7 and the sectional aerosol module - SALSA, to study how the simulated impacts of geoengineering depends on the injection strategy and magnitude and how these results depend on the aerosol microphysical module used.

## 2 Models and Simulations

### 2.1 Aerosol-climate model ECHAM-HAMMOZ

Simulations were done with aerosol climate model ECHAM-HAMMOZ (ECHAM6.3-HAM2.3-MOZ1.0) (Zhang et al. (2012);

Kokkola et al. (2018); Schultz et al. (2018); Tegen et al. (2019)). The host model is the general circulation model ECHAM6.3 (Stevens et al. (2013)). Simulations were performed with a T63L95 resolution which corresponds approximately to a 1.9° × 1.9° horizontal grid. The atmosphere was divided into 95 vertical levels reaching up to ∼ 80 km. This resolution enables to resolve the quasi-biennial oscillation (QBO) in the tropical stratosphere which has an impact on the transport of the stratospheric aerosols.





The aerosol module HAM is interactively coupled to ECHAM and its radiation module (Tegen et al. (2019)). HAM calculates emissions, removal, the radiative properties for the major global aerosol compounds of sulfate, organic carbon, black carbon, sea salt, and mineral dust. It includes gas and liquid phase chemistry of sulfur. ECHAM-HAMMOZ also includes the chemistry model MOZART. Using this model configuration would allow online calculation of ozone and the hydroxyl radical (OH) which is the main oxidizing agent of $SO_2$. However, this model configuration is computationally heavy and would triple the

computational time and its impact on stratospheric sulfur field was relatively small compared to the impact of microphysical processes in our test simulations (not shown). This is because in the case of the continuous injection rates mean that only a fraction is injected each day. Thus stratospheric OH concentrations are not as depleted as in the case of large volcanic eruptions where several megatons of sulfur are dumped during few hour period. However as shown by (Richter et al., 2017) SAI has significant impact on the ozone concentration which further have impact on atmospheric dynamics and injected aerosols.

Nevertheless we did not include MOZART as an active component in our simulations and OH and ozone concentrations were prescribed by a monthly mean climatology. The sea surface temperature, sea ice, as well as the atmospheric GHG concentration and aerosol emissions were fixed to year 2005 levels. The aerosol surface emissions were based on the ACCMIP (Emissions for Atmospheric Chemistry and Climate Model Intercomparison Project) anthropogenic emission inventory. Emissions for sea salt and dust are calculated online.

In this study, when we discuss about "Radiative forcing" we refer to the instantaneous radiative forcing, which is calculated by a double radiation call with and without aerosols and as a difference between specific SAI experiment and control simulation.

The microphysical processes of nucleation, condensation, coagulation, and hydration were simulated by the microphysical module. For this, ECHAM-HAMMOZ has two options: SALSA, where aerosols are represented by size bins of fixed width, and M7, where aerosols are represented using lognormal modes. Both modules have been shown to simulate the stratospheric

aerosol loads and radiative properties consistently compared to the observations of the Mt. Pinatubo 1991 eruption (Kokkola et al. (2018)). However when using M7 this requires slight changes in model configuration to improve representation of the stratospheric aerosols (Kokkola et al., 2009).

### 2.1.1    Sectional aerosol module - SALSA

SALSA describes aerosols using 10 size bins in size space. The seven largest bins are represented separately for soluble

and insoluble material. A detailed description of the SALSA is found in Kokkola et al. (2018). For this study, we made one change to the definition of size bins in the the default setup in SALSA (Kokkola et al., 2018). SALSA bins are divided into subregions, where the first subregion covers the three smallest bins and the second subregion covers the rest of the seven larger bins. The particle size distribution (i.e moving particles from one size bin to another) is updated (at each time-step) based on the mean volume of the bin assuming that aerosols in the bin are evenly distributed and on the actual mean volume of the

particle population (calculated based on the mass and number of aerosols) in the corresponding bin, after the microphysical processes have been calculated. If the actual mean volume of the particle population is larger than the mean volume of the monodisperse size bin, a certain part of the aerosol population is moved to the next bin. This method is called hybrid bin (Young (01 Oct. 1974),Chen and Lamb (15 Sep. 1994)). The new particle formation scheme uses the Kerminen and Kulmala





(2002) J3 parametrization, which produces aerosols with a diameter of 3 nm. 3 nm diameter is also the lowest bound for the SALSA size distribution in the standard setup. Thus, the produced 3 nm particles are smaller than the volume mean diameter of the smallest bin in the default configuration. In the case where new particle formation is efficient, the produced 3 nm particles might keep the actual mean diameter of the smallest bin low. This prevents particles in the bin to be moved to the next bin. This led to an very high number concentration in the smallest bin in our first preliminary simulations. This was solved by changing the size range of first subregion (three smallest bins) so that the volume mean diameter of smallest bin was the same

as the diameter of the newly formed particles (3 nm). This lead to a smaller number concentration in the smallest size bin and a clearly higher concentration in second smallest bin.

### 2.1.2 Modal aerosol module - M7

In M7, aerosols are represented using a superposition of seven log-normal modes, 4 for soluble and 3 for insoluble material. A detailed description of M7 is found in Vignati et al. (2004) and M7-configuration of ECHAM-HAMMOZ in Tegen et al.

(2019). The original modal setup is designed to represent tropospheric conditions which is not representative for cases where the lifetime of particles is long (>months) e.g in case of the stratospheric sulfate aerosol intervention (Kokkola et al., 2009). Similarly to Niemeier and Timmreck (2015), we modify the setup of the modes so that the coarse mode is made narrower than in the standard setup (using standard deviation $\sigma_{CS}$ = 1.2 instead of 2.0) and the threshold radius when aerosols from the accumulation mode are transferred to the coarse mode is changed from 0.5 $\mu m$ to 0.2 $\mu m$. In case of high sulfur concentrations,

a 2.0 $\mu m$ coarse mode width has shown to lead to a tail of large particles (Kokkola et al., 2009). Based on Kokkola et al. (2009) this caused an overestimation of the effective radius of the coarse mode, when compared to the highly resolved particle spectrum reference model, and thus increased sedimentation velocity and reduced residence time of aerosols. Our model setup does not include all modifications done in (Niemeier and Timmreck, 2015) e.g. the simple stratospheric sulfur scheme. Even though the mode setup of the model was modified to represent well the stratospheric aerosol at the expense of representation of the

tropospheric aerosols (especially sea salt and dust), we also include all anthropogenic emissions and natural surface emission.

### 2.2 Scenarios

Studied scenarios are listed in Table 1 and were simulated with both SALSA and M7 aerosol modules. In addition, the control (*CTRL*) simulation without SAI was simulated with both microphysics models. In our *Baseline* scenario, sulfur was injected at 20-22 km altitude and a band between the latitudes 10° N and 10° S. To study the sensitivity of radiative forcing to the

magnitude of the injection, the yearly injection rates of 1, 2, 5, 10, 20, 50, 100 Tg(S) were used. In addition, we simulated eight sensitivity scenarios with alternative injection strategies. These were done for injection rates 2, 5, 20 and 50 Tg(S)yr$^{-1}$. Scenarios *Narrow* and *Wide* were simulated to study the impacts of shrinking or widening the injection area. Scenarios *Low* and *High* were done to study the dependency of radiative forcing on altitude. We also simulated two scenarios, where injections are concentrated on the certain times of the year and not continuously over a year. In this case the concentration of injected

sulfur is larger compared to the continuous injections with same yearly injection rate, but on the other hand, there are times when injections are suspended and sulfate aerosol field is not under new injections. This might affect on the size distribution





**Table 1.** Simulated scenarios - Linestyles and markers used to indicate scenarios in the figures are shown beside the corresponding scenario name. SALSA *Baseline* results are colored in blueish and M7 results by reddish colors in sect. 3.1 and in black in sect. 3.3

| Baseline | | |
|---|---|---|
| *SRM1-100* | | continuous 1, 2, 5, 10, 20, 50 and 100 Tg(S)/yr SO$_2$ injections to 10° N - 10° S and 20-22 km |
| **Sensitivity** | | **2, 5, 20 and 50 Tg(S)/yr SO$_2$ injections** |
| *Narrow* | | 1.9° N - 1.9° S and 21 km, continuous |
| *Wide* | | 30° N - 30° S and 20-22 km, continuous |
| *High* | | 10° N - 10° S and 22-24 km, continuous |
| *Low* | | 10° N - 10° S and 18-20 km, continuous |
| *Pulse2m* | | 10° N - 10° S and 20-22 km, injection every two months |
| *Pulse6m* | | 10° N - 10° S and 20-22 km, injection every six months |
| *Onegb* | | Injections to two gridbox at the Equator, 21 km, continuous |
| *Seasonal* | | 20° wide area varies seasonally between 40° N and 40° S, north most position at May, 20-22 km, continuous |

of the stratospheric aerosols. In both of these scenarios the length of the one injection period is one month. In *Pulse2m*, sulfur was injected in during every two months starting from January. In *Pulse6m* scenario, sulfur was injected during two months per year, January and July.

In *Seasonal* scenario, 20° wide injection area is varied throughout the year similarly to Laakso et al. (2017). The northern-most position (40° N - 20° N) of injection area is in May and the southernmost position (20° S - 40° S) is in November. Note that results of scenarios *Seasonal* or *Pulse6m* are also sensitive to the timing of injections which, however, are not studied here. One sensitivity scenario was *Onegb*. There, sulfur was injected only in one meridional grid box instead of a band over all longitudes, and to have a symmetry between hemispheres, and regardless of the name of the scenario, to two grid boxes closest to the Equator. These two grid boxes were located in prime meridian.

Simulations were run over a 10 year period which included a 3-year spin-up period. Thus a 7 year period was used in the analysis. The period length of 7 years was chosen because it covers roughly three full QBO cycles ( 3 x 28 months = 7 years).

# 3 Results

## 3.1 *Baseline* scenarios - Sensitivity to magnitude of injections

### 3.1.1 The dependency of radiative forcing on the amount of the injected sulfur

Figure 1 shows the dependency of the global mean All-sky SW and LW radiative forcing, the stratospheric sulfur burden, and the effective radii of aerosols on the magnitude of sulfur injection in the scenario *Baseline*. The results of SALSA are shown





by reddish lines and markers, M7 results are indicated by blueish color. The SW radiative forcing increased and the forcing efficiency decreased sub-linearly with the injection rate in both models. However, the increase in LW forcing was rather linear

as a function of the magnitude of injected sulfur (Fig. 1b). This was consistent between models. Overall, because SW forcing was significantly larger than LW forcing, the net total forcing was always more negative (stronger cooling effect). However, in the case of stronger injections (> 5 Tg(S)yr$^{-1}$) LW forcing contribution to the total forcing becomes relatively higher especially in simulations with M7. Thus, for example the change in the total forcing in simulations with M7 was rather small (-2.09 Wm$^{-2}$) even though the amount of injected sulfur was doubled from 50 to 100 Tg(S) (Fig. 1c). Several studies have shown

that a stronger sulfur injection will lead to relatively larger aerosols (Heckendorn et al. (2009), Pierce et al. (2010), Niemeier et al. (2011), Laakso et al. (2016)). This happens also here regardless of how the aerosol microphysics is modelled. This is supported by Fig. 1d which shows that the area-weighted mean stratospheric effective radius was increased with increasing injections.

Even though the same qualitative conclusions about the behaviour of the efficiency of SAI as a function of the amount

of injected sulfur can be drawn based on both SALSA and M7 microphysical modules, the quantitative results between the models were significantly different. The SW radiative forcing was 45-85% higher in SALSA than in M7. In other hand the LW radiative forcing was 32-67% higher in M7 than in SALSA. As impact on net radiation is opposite between the SW and LW radiative forcing of aerosols, this led to an even larger difference in the total net radiative forcing between models as can be seen in Fig. 1c. With 1 Tg(S)yr$^{-1}$ and 2 Tg(S)yr$^{-1}$ injection rates, the total radiative forcing was 88% and 117% higher

in SALSA than in M7. In the case of higher magnitude injections (5-100 Tg(S)yr$^{-1}$), the net radiative forcing was 137-154% based on simulations with SALSA than with M7. Thus, the efficiency of stratospheric sulfur geoengineering was significantly dependent on the aerosol module used.

The net radiative forcing in our M7 simulations was in very good agreement with the results of Niemeier and Timmreck (2015) which are indicated with black dashed line in Fig. 1c. This was at least partly a coincidence, even though M7 with

a similar mode setup was used here as in Niemeier and Timmreck (2015). Niemeier and Timmreck (2015) used ECHAM5-HAM instead of ECHAM6.3-HAM2.3 which was the case in our study. Niemeier and Timmreck (2015) simulations were done using 39 model vertical levels instead of 95 and the injection altitude was 19 km which was thus a lower altitude than in our simulations. In addition, the sulfur was injected to the one grid box instead of injecting along the Equator which was the case in our simulations. If the SALSA results presented here are compared with the results of Kleinschmitt et al. (2018),

where a sectional aerosol module with 36 size bins between dry radii 1nm and 3.3nm was used, we see a significant difference especially in the LW radiation response. In the case of stronger than 5 Tg(S)$^{-1}$ injection rates, our simulation showed the LW forcing efficiency to be lower than 0.1 Wm$^{-2}$/Tg(S)yr$^{-1}$ while in Kleinschmitt et al. (2018) it was approximately 0.3 Wm$^{-2}$/Tg(S)yr$^{-1}$. This means that the LW forcing was more than two times larger in Kleinschmitt et al. (2018) than in our simulations. Ozone were prescribed in Kleinschmitt et al. (2018) as also in this study and thus different responses in LW forcing

cannot be explained by different response of ozone on SAI. The SW forcing efficiency was slightly larger in SALSA simulations with the 1-2 Tg(S)yr$^{-1}$ but with stronger injection rates, the results are consistent with those of Kleinschmitt et al. (2018). The dry effective radii of stratospheric aerosols with different injection magnitudes were nearly identical between the studies. This





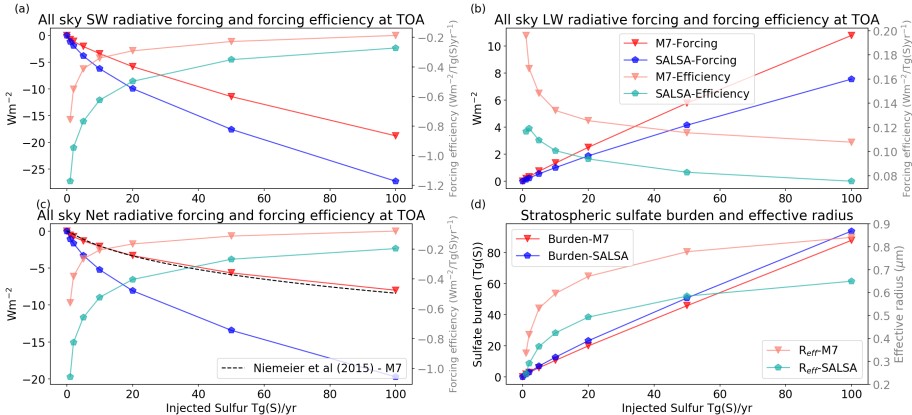

**Figure 1.** Global mean all-sky a) shortwave, b) longwave and c) total (net) radiative forcing and d) stratospheric sulfate burden and the mean effective radius as a function of injection rate. M7 results are shown by red lines and markers and SALSA results are indicated by blue color.

indicates that differences in radiative forcings between the studies are caused by differences in the LW radiation transfer, i.e in using a different radiative transfer scheme or differences in the aerosol optical properties in LW radiation calculations. In addition, in Kleinschmitt et al. (2018) radiative properties of aerosols were calculated from a prescribed chemical composition consisting of 75% $H_2SO_4$, while in SALSA volume-weighted average of the refractive indices of individual compounds is used. However, it can also be that the size-distribution of aerosols were different regardless of having consistent effective radii, or how aerosols are spatially distributed in atmosphere.

Despite the fact that the radiative forcing was significantly different here between M7 and SALSA, the stratospheric sulfur burdens were only 3-19% higher in SALSA than M7. This indicates that there were fewer and larger sulfate particles in M7 simulations than in SALSA. This conclusion is justified by examining the stratospheric mean effective radii (the light blue and red lines) in Fig. 1d. To analyze the aerosol size in more detail, the number size distribution along the Equator at 20-22 km of altitude in M7 and SALSA simulations for 5 and 50 Tg(S)yr$^{-1}$ are shown in Fig. 2. The total number concentrations were larger in SALSA than in M7 in all size classes except in particles with diameter larger than 0.7 $\mu m$. Note that in the case of the 50 Tg injections, a part of the aerosols is present in the largest bin (of size range 1.7-4.12 um), whose upper size limit goes beyond the coarse mode in M7. However, the actual mean aerosol size for that bin (blue circle in the bin) is closer to its lower limit, unlike the other bins.

To understand the link between the size distribution and the radiative forcing we reproduced an indicator for size range where the backscattering efficiency is highest, similarly to Figure 5 of Vattioni et al. (2019). Defined size range is based on Dykema et al. (2016). The size range is shown as a light green shaded area in Fig. 2. The magenta line shows the dependency of the LW absorption for radiation with wavelength of 8000 nm to the size of sulfate aerosol calculated using the SALSA radiation module for aerosols (absorption is shown here as an unitless quantity). In SALSA, the aerosol number concentration was much higher than in M7 over the the size range of highest backscattering. On the other hand, high number concentration at the largest





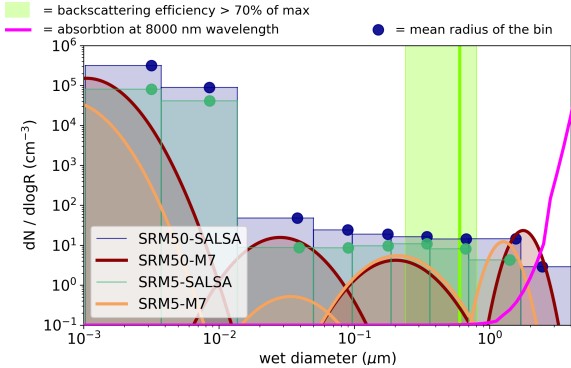

**Figure 2.** Aerosol number size distribution at the Equator and at the 20-22 km altitude in scenarios SRM5 and SRM50 simulated with M7 and SALSA. Dots on the top of the SALSA size bins show the mean diameter of that bin. The light green line is reproduced from Figure 5 of Vattioni et al. (2019) and shows the size for which the back scattering is maximized. Shaded area indicates radius where aerosol backscattering is 70 % of the maximum Dykema et al. (2016). Magenta line (here shown as a unitless quantity) shows the relative dependence of absorption of 8000 nm wavelength on the (dry) diameter of the sulfate aerosols, based on the radiation module of SALSA.

size range (>0.7 $\mu m$) in M7 caused stronger LW radiative forcing than the LW forcing calculated for the SALSA simulated
size distribution. Figure 2 shows clearly why the net radiative forcing increases (becomes more negative) faster in SALSA than M7. When the injection rate was increased from 5 to 50 Tg(S)yr$^{-1}$, the number concentration was increased in all size bins in SALSA. However, in M7 changes in the accumulation mode were small. On the other hand, the number concentration of the coarse mode was increased and it grew in size. As seen in Fig. 2, this change is critical for LW radiative forcing because the absorption efficiency increases strongly with the aerosol size when the aerosol diameter is larger than $1\mu m$. Compared to the
aerosol size distribution in Niemeier and Timmreck (2015), it was considerably different in our M7 simulations. The number concentrations of Aitken and Accumulation modes were much larger in Niemeier and Timmreck (2015) and the amount of accumulation aerosols was increasing with larger injections. These differences are probably explained by the different injection strategy. As we will show in sect. 3.3.2 the scenario where sulfur is injected to one model grid box as in Niemeier and Timmreck (2015) results in the more consistent aerosol size distribution (Fig. S5b in supplement) with Niemeier and Timmreck (2015).
In addition, a different version of the same GCM and a different resolution was used here, resulting in different atmospheric dynamics.

### 3.1.2 Dependency of zonal distribution of radiative forcing on the amount of the injected sulfur

Figure 3 shows the zonal distribution of the relative clear sky SW and LW forcings (i.e. zonal forcing/global mean forcing). The maximum of the zonal forcing was concentrated on latitude bands within the injection region (the shaded area in Fig. 3) over
the Equator in both models and regardless of the magnitude of injections. There were also two local maxima: both 50° N and 50° S latitudes especially in the zonal SW radiative forcing. When the injection rate was increased, the relative radiative forcing at high latitudes decreased and the above-mentioned local maximas of the SW radiative forcing at 50° latitudes had moved





towards low latitudes. Consequently, the relative radiative forcing increased in the tropics and subtropics while it decreased over higher latitudes. This was consistent between the models. There are two explanation for this: 1) when the amount of sulfur

was increased, aerosols became relatively larger thus having higher gravitation settling velocity and 2) as it has shown, injected sulfate aerosols can strengthening the polar vortex with higher tropical stratospheric heating, preventing particles from reaching the polar stratosphere (Visioni et al. (2020a) see also sect. 3.4). Thus, less sulfate aerosols were transferred to high latitudes before they were removed from the atmosphere. The variation seen in the LW forcing with low injection rates (1-2 $\mathrm{Tg(S)yr^{-1}}$) is caused by background aerosols and the variation in the emitted LW radiation from the atmosphere and the surface due to

land temperature adjustments.

The zonal mean effective radii were notably different between the models (Fig. 4). The impact of the injection area on the aerosol effective radii is clearly seen in SALSA between latitudes 10° N to 10° S, where the zonal mean effective radius over the injection band was smaller than over the higher latitudes. This indicates that continuous injections were resulting in continuous new particle formation in SALSA. When particles were transferred out from the injection area, the effective radius

began to rise due to particle growth by coagulation and condensation, while less new particles were produced by nucleation. In M7 simulations, the effective radii were much larger over the tropics than over high latitudes. This indicates that more of the injected sulfur has condensed on pre-existing particles rather than forming new particles inside of injection area.

Figure 5 shows the aerosol number size distribution over the latitudes 20 °N and 50 °N. When the aerosol plume moved towards high latitudes, the number of the Aitken and accumulation mode aerosols began to increase and coarse mode to

decrease compared to the size distribution over the Equator. Thus, the aerosol effective radii over latitudes 20 °N and 50 °N were much smaller compared to effective radius at the Equator. In SALSA, the number size distribution at 20 °N and 50 °N is three modal in shape while at the Equator, excluding two smallest bins, it was relatively monodisperse. At the 50 °N latitude the number size distribution and effective radius were more consistent between models than closer to the injection area. However, there is still a gap between accumulation and coarse modes in M7.

In addition to radiative impacts, the size of the aerosols affects where and how fast particles are removed from the atmosphere. As Fig. S1 in the supplement shows, deposition at the tropics was much faster (32% in 50 $\mathrm{Tg(S)yr^{-1}}$ injection rate) in M7 than in SALSA while the deposition outside the tropics was slower (roughly 10% in SRM50). This conclusions holds regardless of the amount of injected sulfur. Simulations were too short to make a statistically significant detailed analysis of geographical distribution of deposition in the case of lower than 10 $\mathrm{Tg(S)yr^{-1}}$ injections due to the deposition of the background

sulfate aerosol. However, an estimation could be done based on the highest injection rates even though they present probably unrealistic extreme scenarios. Figure S2 in supplementary material shows that, in the case of 50 $\mathrm{Tg(S)yr^{-1}}$ injection rate, the deposition of sulfate is clearly faster in SALSA than M7 e.g. over Europe and United States. An enhanced sulfate deposition due to the SAI might offset or even exceed impacts of reduction of anthropogenic $SO_2$ emissions which might have negative impact on ecosystems in these regions Visioni et al. (2020b).





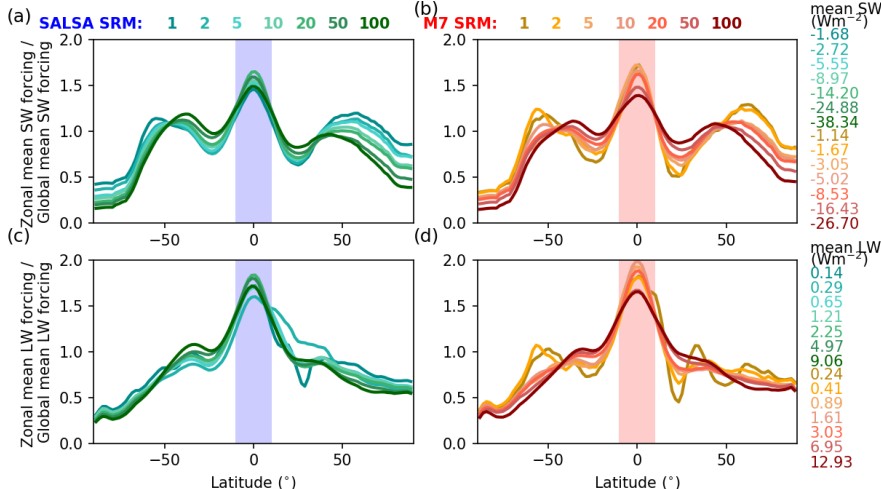

**Figure 3.** Relative zonal distribution of a-b) shortwave and c-d) longwave clear sky radiative forcing. Zonal mean forcing in each latitude band is divided by global mean radiative forcing of corresponding scenario which are shown in the legend right side of the figure. Blue and red shaded areas show latitudes where sulfur is injected

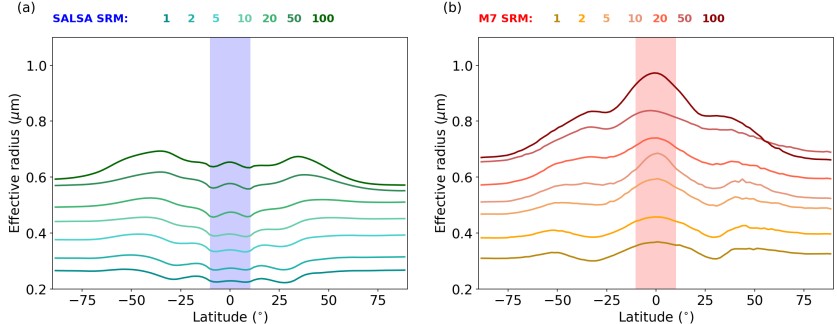

**Figure 4.** Dependence of the zonal mean effective radius of stratospheric aerosols on the magnitude of sulfur injections simulated with a) SALSA and b) M7. Blue and red shaded areas show latitudes where sulfur is injected

## 3.2 Analysing the causes for differences in results between SALSA and M7

Based on Fig. 4, there was a significant difference in the evolution of aerosols within the injection band between the two models. In there, large amounts of gaseous $H_2SO_4$ is constantly produced from continuous $SO_2$ injection and oxidation with OH. In the stratosphere the conditions are favorable for new particle formation through $H_2O$–$H_2SO_4$ binary nucleation but on the other hand, there is also a large amount of pre-existing sulfate aerosols to which gaseous $H_2SO_4$ can condensate. These two processes compete for available gaseous $H_2SO_4$ and solving them simultaneously in the model is challenging especially when sulfur concentration is high, and can lead to large biases (Kokkola et al., 2009; Wan et al., 2013). SALSA and M7 allocate





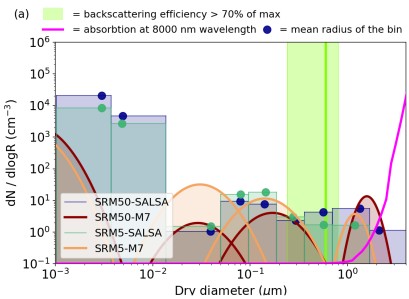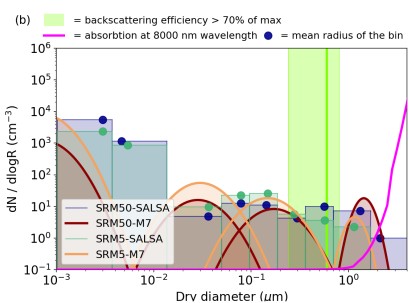

**Figure 5.** Aerosol number size distribution at the a) 20 °N and 18-20 km altitude and b) 50 °N latitudes and 12-15 km altitude in scenarios SRM5 and SRM50 simulated with M7 and SALSA. Dots on the top of the SALSA size bins are showing a mean diameter of the bin. Light green line is reproduced from from figure of Vattioni et al (2019) and shows to size of the maximum back scattering and shaded area indicates radius where aerosol backscattering is 70 % of maximum Dykema et al. (2016). Magenta line shows (unitless) relative dependence of absorption of 8000nm wavelength on the (dry) diameter of the sulfate aerosols, based on the radiation module of SALSA.

the amount of sulfuric acid partitioning from gas to particles between new particle formation and condensation differently. Based on the results, in SALSA most of the gaseous sulfate is partitioned into new particle formation, while few goes into condensation. In addition, in SALSA there is a high number of particles smaller than 10 nm which also prevents gaseous

sulfuric acid from condensing to the larger particles. On the other hand, the number concentration of particles smaller than 10 nm at the Equator in M7 was only 34% of the number concentration in SALSA and, as Fig. 4 shows, in M7 the effective radius was larger inside the injection region compared to latitudes where injections did not take place. This indicates that new particle formation is much lower in M7 than SALSA and that sulfate gas condensates onto existing particles inside the injection area, which results in a larger number and size of coarse mode particles compared to latitudes where no injection is

taking place.

In M7, the coupling of nucleation and condensation is done by a two-step time integration scheme proposed by Kokkola et al. (2009). Based on Wan et al. (2013) this has shown to cause negative bias for the nucleation sink and positive bias for the condensation rate. In SALSA, the operator splitting technique ((Jacobson, 2002)) is used (Bergman et al., 2012). In this method, nucleation rate is added to the condensation mass transfer rate in the first size bin. Based on the test simulations with

the box model (not shown), when nucleation takes place, it overruns condensation. This conclusion is supported by Fig. 4. It is not clear if this is caused by overestimation of the nucleation rate, underestimation of the condensation rate, or the method used for solving nucleation and condensation simultaneously. However, to study how large an impact from the competition between nucleation and condensation have on different results between models we did additional simulations where the competition is excluded in both models. These results are shown in Appendix. In these simulations, nucleation was switched off and 25% of

injected sulfur mass was assumed to be primary particles 3 nm in diameter while the rest of the sulfur was injected as SO$_2$. These simulations showed that excluding nucleation brought the global mean net radiative forcing results between models closer to each other. However, there is not a large difference in aerosol number size distribution for particles larger than 0.1 μm aerosols between *Baseline* simulations and modified simulations without nucleation (see appendix Fig. A3). Thus the most of





the differences in raditive forcings between models are not caused by differences in calculating competition between nucleation and condensation.

In M7, aerosol sizes are more restricted to the definition of the modes than in SALSA with bins. In M7, the mode widths are fixed and the radius of the each mode have fixed low and high limits. E.g accumulation mode (second largest mode) has low and high radius limits of 0.05 and 0.2 μm respectively. These limits define average mass of the mode. If the average mass of the particles in the mode is exceeding the defined average mass based on lower and upper limits, the transfer of number and mass is done to the next mode. The impact of this can be seen e.g in Fig. 2. In simulations with both 5 and 50 Tg(S)$^{-1}$ injection rates, the average mass of the accumulation mode (third mode from the left) was close to the upper limit of mode and thus it cannot grow by condensation or coagulation because gained extra mass is always transferred to coarse mode which also decrease the number of accumulation mode particles. The number concentration of accumulation mode can only increase by coagulation of two smaller particles or through the growth of the Aitken mode. However because coagulation between larger and smaller aerosols are more efficient than between small aerosols, and because the number of coarse mode particles is relatively high, the Aitken mode cannot compete with the Coarse mode as an coagulation sink for nucleation and Aitken mode aerosols. This lowers growing of the Aitken mode aerosols to the accumulation mode size by coagulation. This creates self-reinforcing loop when number and mass of coarse mode increases. Because size range of accumulation mode is restricted by upper limit while coarse mode aerosols is getting larger with the larger injection, there is a gap in size distribution between these modes where the aerosol number concentration is low (Fig. 2). Coincidentally this gap is located at the size range of the largest backscattering efficiency, which is indicated by green shaded area in Fig. 2. Thus the modal setup of M7 causes a numerical limitation for particle size distribution which, in this case, have an impact on efficiency of SAI. Note that based on the earlier SAI simulations by M7 Niemeier and Timmreck (2015), the threshold radius when aerosols from accumulation mode is transferred to coarse was set to 0.2 $\mu m$ (0.5 $\mu m$ in standard setup).

Because Pinatubo is the only large volcanic eruption, which has taken place during time, when proper observations about radiative properties of stratospheric aerosols are available, it has often used as a test case for models capability to simulate stratospheric aerosols (eg. (English et al., 2013; Mills et al., 2017; Niemeier et al., 2009; Laakso et al., 2016; Sukhodolov et al., 2018)). However, based on our results, it probably does not give a reliable picture of the model capability to simulate stratospheric sulfur injections for SRM. For example both models used here have shown to represent effective radius of sulfate aerosols and the burden of sulfur after Pinatubo eruption relatively well (Niemeier et al., 2009; Laakso et al., 2016), and especially results were shown to be very good agreement with each other (see Figure 16 in Kokkola et al. (2018)). Still the model responses in case of SAI are significantly different. This is because background conditions in case of volcanic eruption and continuous sulfur injections are significantly different. While in case of the volcanic eruption, sulfur is erupted in a relatively particle free stratosphere, in case of SAI, sulfur is injected into existing particle field in the stratosphere. In the former case competition between nucleation and condensation does not have as large role as in case of SAI.



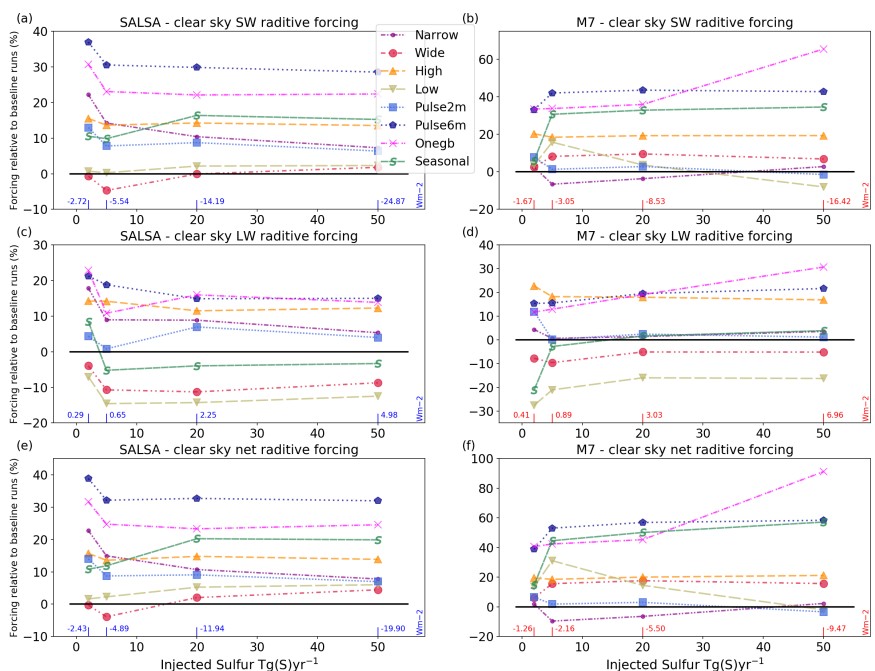

**Figure 6.** Relative global mean clear sky SW (a-b), LW (c-d) and net radiative forcing in sensitivity scenarios compared to *Baseline* scenario with corresponding sulfur injection rate. *Baseline* values are shown at the bottom of each panel. SALSA results are shown in the left and M7 in the right panels. Note the different y-axes scale between the panels.

## 3.3 Sensitivity scenarios - Sensitivity to injection strategy

In this section we investigate the impact of various injection strategies on the geoengineering efficiency and the zonal distribution of the radiative forcing and how the responses depend on the model used. The descriptions of the sensitivity scenarios are found in Table 1. Fig. 6 shows the relative difference in the global mean clear sky SW, LW, and net radiative forcing compared to the *Baseline* scenario for a corresponding injection magnitude. The relative sulfate burdens compared to the *Baseline* scenario and the effective radii of stratospheric aerosols are shown in Fig. 7, while the tabulated values are given in the supplement (Tab S1-2). The zonal mean net clear-sky radiative forcing is shown in Fig. 8. The zonal mean effective radius of stratospheric aerosols is shown in Fig. 9. We show here the clear-sky radiative forcing instead of all-sky, because in clear sky conditions it is more straightforward to compare the radiative forcings to the aerosols size as clouds do not affect the results. Figures for all-sky radiative forcings and tabulated absolute values of clear sky and all sky SW, LW and net radiative forcings are shown in the supplement (Fig S3).





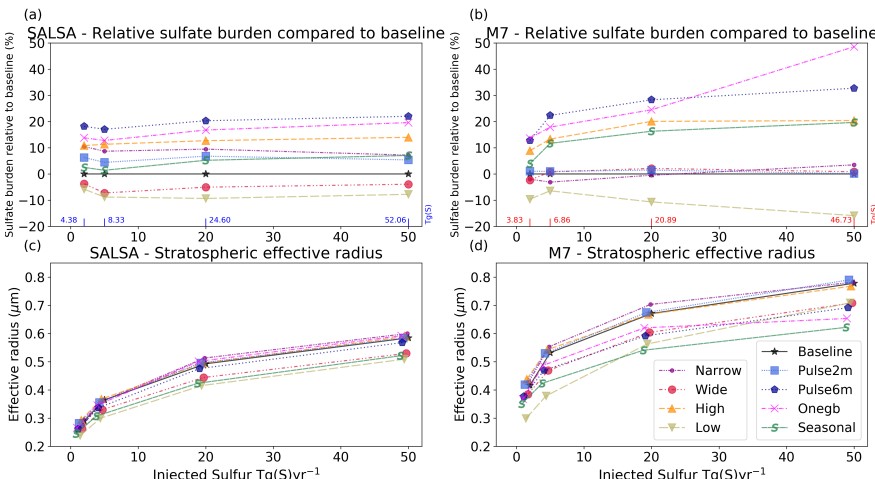

**Figure 7.** a-b) Relative stratospheric sulfur burden in sensitivity scenarios compared to *Baseline* scenario with corresponding sulfur injection rate. c-d) Global mean effective radius of stratospheric aerosols

### 3.3.1 Sensitivity to the width of the injection area

To investigate the sensitivity of radiative impacts of geoengineering to the width of the injection area, we studied injection of sulfur to bands with widths of 4° (*Narrow*), 20 ° (*Baseline*) and 60 ° (*Wide*) over the Equator. Responses of the radiative

forcing to the widening of the injection area from 20 to 60 ° were similar in both models. The effective radii of stratospheric aerosols were smaller and the LW radiative forcing was lower in both models compared to the *Baseline* scenario. Widening the injection area affects the radiative forcing in two ways. It decreases the mean sulfate concentration over the tropics and it results in relatively more and smaller particles. These effects are shown as smaller effective radii (Fig. 7) and lower absorption of LW radiation by the smaller particles. On the other hand, injecting sulfur farther from the Equator, where the solar intensity

is largest on average, decreases the potential of aerosols to scatter radiation. Widening the injection area decreases also the lifetime of aerosols because some of the aerosols are injected closer to high latitudes where they are removed faster than over the low latitudes. In SALSA, where the condensation on the existing particles is weaker than in M7, concentrating sulfur injection to wider area does not matter as much as in M7 in microphysical sense, because nucleation happens at the expense of condensation even in high sulfur concentrations. Thus lifetime of aerosols in SALSA is reduced due to the more efficient

removal when injected to higher latitudes compared to *Baseline*. In M7, there is not large difference in lifetime of aerosols between *Baseline* and *Wide* scenarios. Overall, the global mean total (SW+LW) radiative forcing was roughly 20% larger than in the *Baseline* scenario when simulated with M7 while with SALSA, the difference was between ±4% depending on the injection rate.

Wider injection area decreases the radiative forcing over the tropics while increasing it at higher latitudes compared to

*Baseline* scenario (Fig. 8). However, the difference in radiative forcings at the tropics with 50 Tg(S)yr$^{-1}$ injection rate between



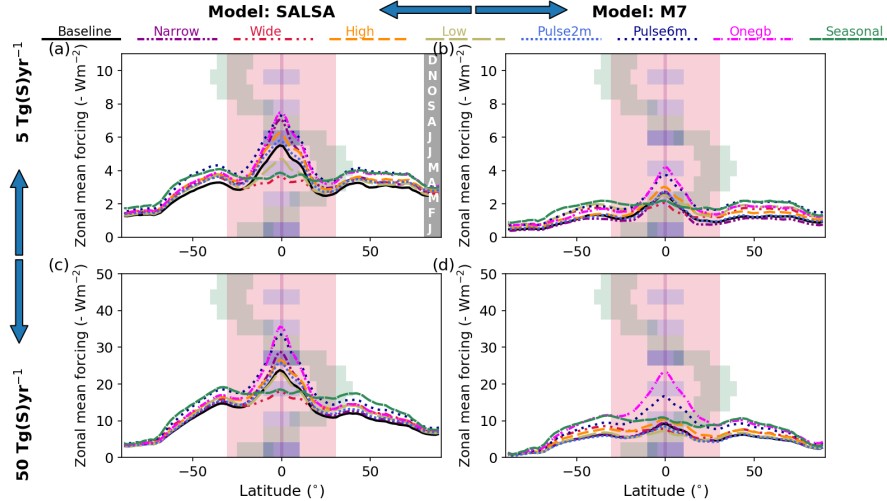

**Figure 8.** Zonal mean clear sky net forcing with a-b) 5 and c-d) 50 Tg(S)yr$^{-1}$ injection rates in simulated sensitivity scenarios. SALSA results are shown in the left and M7 results on the right panels. Note that y-axes scale shows negative (cooling) values. Shaded areas show latitudes of injection area at the time of year (y-axes, month shown in right edge of panel a)) in each injection scenario. Latitudes of injection area are same in *Baseline*, *High* and *Low* scenario (10° N - 10° S).

scenarios *Baseline* and *Wide* was rather small in M7(<10%). In the *Baseline* scenario, higher injection concentration due to the narrower injection area (10° N - 10° S) enhances the effective radius of stratospheric particles in M7 to be larger than 0.8 $\mu m$ which is not an optimal size for scattering radiation (Fig. 9).

As expected, a narrower injection area (two model grid boxes over the Equator) (*Narrow*-scenario) led to larger effective
radii of stratospheric aerosols in both models compared to the *Baseline* scenario. However, sulfur lifetime increases as sulfur is injected into stratospheric tropical pipe where uplifting is even further increased due to heating of the atmosphere caused by aerosols (see sect. 3.4). As in the case of scenario *Wide*, the impact of the locally larger injection rate does not increase effective radii of aerosols in SALSA as much as in M7 and the lifetime of particles was 10% longer in scenario *Narrow* than in scenario *Baseline* due to the impact of atmospheric circulation. Thus, the radiative forcing of injection scenario *Narrow* was larger than
in the *Baseline* scenario and it decreased gradually from 23% to 8% when the magnitude of sulfur injections was increased from 2 to 50 Tg(S)/yr, based on SALSA simulations. Simulations with M7 show that the *Narrow* injection strategy does not affect significantly the lifetime of aerosols and the net radiative forcing in the *Narrow* scenario was of the same magnitude or slightly lower than in the *Baseline* scenario in M7.

### 3.3.2 Injecting to one grid box instead of a band over longitudes

In scenario *Onegb*, the injection area in *Narrow* was shrunk also in the meridional direction, reducing it to one gridbox. This further increased sulfur injection rate in the injection area as sulfur injections are concentrated on the smaller region. However,



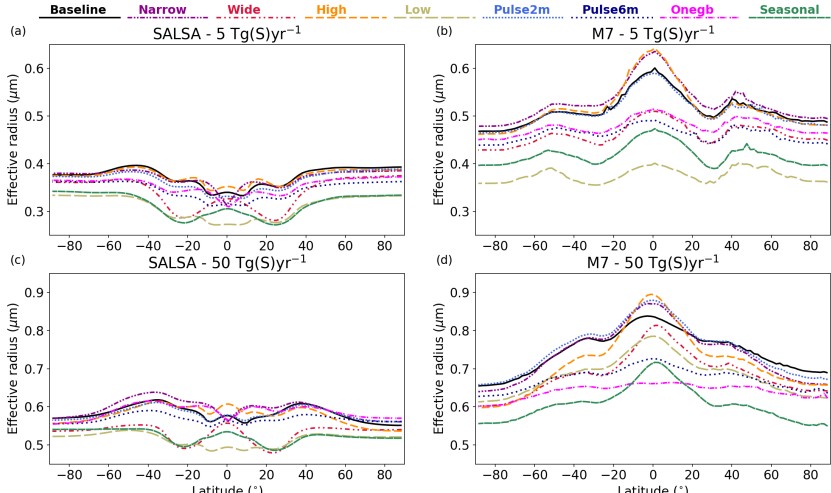

**Figure 9.** Zonal mean effective radius of stratospheric aerosols in studied sensitivity scenarios for 5 (a and b) and 50 (c and d) Tg(S)yr$^{-1}$ injection rates simulated with SALSA (left panels) and M7 (right panels).

as sulfur is injected longitudinally to area which is only one model grid box wide, less aerosols over the Equator are under continuous injection. Even though sulfur is mixing relatively fast over longitudes, the available gaseous sulfate for condensation or nucleation is localized nearby the injection area in the *Onegb* scenario.

Based on simulations with SALSA, scenario *Onegb* was the second most efficient of the studied SAI strategy, regardless of the magnitude of injections. The mean net clear-sky radiative forcing in the *Onegb* scenario was 23-32% larger than in the *Baseline* scenario, depending on the injection rate. Results of the *Onegb* scenario with M7 showed significantly different behaviour when increasing injection rate compared to the other scenarios and even in the same scenario using SALSA. While the clear sky global mean net radiative forcing was roughly 40% larger compared to the *Baseline* scenario with injection rate

of 2, 5 and 20 Tg(S)y$^{-1}$, it was 91% larger in the case of injection rate of 50 Tg(S)y$^{-1}$. To study this in more detail, additional simulations of *Onegb* injection scenario were simulated with M7 and with 10, 30, 40, 70 and 100 Tg(S)y$^{-1}$ injection rates. The Global mean SW radiative forcing, forcing efficiency, and the lifetime and effective radius of stratospheric aerosols from these simulations are shown in Fig. S4 in the supplement. All simulations with SALSA and all other scenarios with M7 were showing the following: the effective radius increases and the SW radiative forcing efficiency and lifetime of aerosols decrease

with an increasing injection rate. However in the *Onegb* scenario with M7, the lifetime of aerosols increased with increasing injection rate when injection rates were larger than 20 Tg(S)y$^{-1}$. In addition, SW forcing efficiency did not decreased and the effective radius of stratospheric aerosols did not increased similarly as in simulations with SALSA and all other scenarios when the injection rate was increased from 20 Tg(S)y$^{-1}$.

    A closer look at the aerosol number size distributions over the Equator in *Baseline* and *Onegb* scenarios shows why the

lifetime and SW radiative forcing increased in *Onegb* scenario with increasing injection rate (Fig. S5). In the *Baseline* scenario,





number concentration of accumulation size aerosols decreased whereas number and size range of coarse mode increased with increasing injection rate. This did not happened in *Onegb* scenario, where the number of accumulation model sized aerosols increased and median radius of coarse mode did not growth the same way as in *Baseline* scenario. In addition, when the injection rate exceeded 30 $Tg(S)^{-1}$, the coarse mode shrank with increasing injection rates, which probably explains the

increase in the mean lifetime of aerosols. This also contradicts the results of Niemeier and Timmreck (2015) where injection to the one grid box were simulated.

It is not totally clear what is causing this peculiar behaviour in this one scenario and only when simulated with M7. The scenario, where sulfur is injected to a single grid box, differs from all others in two ways 1) the concentration of injected $SO_2$ is significantly higher compared to scenarios with injections over a whole latitude band and as was pointed out earlier,

this also may lead to a OH limitation for sulfate formation, which is however not simulated with prescribed chemistry, 2) in the *Onegb* scenario, aerosols over the Equator are not under continuous injections except in one model grid box. Inside the injection area concentration of nucleation particles is high and these particles can grow to the size range of Aitken mode by self-coagulation and condensation. This is seen as larger number concentration at the longitude where injections take place (Fig. S6b). If compared for example to the size distribution of *Baseline* scenario (Fig. 2), the number concentration of Aitken

mode aerosols is significantly larger and thus it can also compete more efficiently with coarse mode for available sulfate gas. This results in also larger number of accumulation mode aerosols than in *Baseline* scenario and thus also larger SW radiative forcing. On the other hand, the size of the coarse mode particles is significantly smaller in *Onegb* than in *Baseline* scenario. The mean radius of the coarse mode is affected by several processes: coagulation and condensation on the coarse mode aerosols are increasing the size of the coarse mode, while sedimentation and reallocation of aerosols from accumulation mode to coarse

mode decreases the mean radius. It seems that with a high enough injection rate (more than 30 $Tg(S)yr_{-1}$) the processes contributing to the shrinking of the mode are more efficient resulting in an overall decrease in the size of the coarse mode. In addition, coarse mode particles, which are carried around Equator, are under continuous injections in *Baseline* scenario and growth to larger size through the efficient condensation and they also coagulate efficiently with the formed new particles. In *Onegb* scenario this is not happen as there is significantly less available $H_2SO_4$ outside of injection area (Fig. S6a).

As was mentioned, this peculiar behaviour was not seen in Niemeier and Timmreck (2015) where similar scenario was simulated with the M7 in earlier generation of ECHAM. However, as the atmospheric model, background conditions (e.g surface aerosol emissions) and resolution were different than here, this peculiar behaviour might be somehow related also atmospheric dynamics. This is supported the fact, that in this study *Onegb* scenario was first simulated with the untuned model version and these peculiar results in *Onegb* were not seen in those simulations. Overall this indicates that this unique

behaviour seen in the lifetime of aerosols and radiative forcing in *Onegb* scenario is probably caused by nonlinearities in the microphysical processes and dynamical changes and restriction of modes in aerosol size distribution in M7. This shows that simulating extreme cases, where sulfur concentration is locally large, might lead to peculiar behaviour of a modal model. This should be kept in mind also when simulating e.g supervolcanoes. In any case, together with the significant difference in model responses seen between M7 and SALSA on the SAI, these peculiar results in *Onegb* highlight need for better tools or

observational data to evaluate models.

Zonally, scenario *Onegb* led to the largest radiative forcing at the Equator out of all studied scenarios in both models. Most notably, *Onegb* stands out in simulations of 50 Tg(S)yr$^{-1}$ injection rate with M7, where radiative forcing is 240% larger over tropics than in *Baseline* scenario. In addition, radiative forcing over the tropics in the *Onegb* scenario with M7 was close to the results of the *Baseline* scenario with SALSA even though simulations with SALSA generally showed much larger radiative

forcing. Several studies have shown that offsetting GHG induced global average warming by SRM lead to cooling at the tropics while high latitudes are warmer in case of equatorial injection or in an idealized case, where SRM is imitated by reducing the solar constant (Aswathy et al. (2015), Jones et al. (2016), Kravitz et al. (2016), McCusker et al. (01 May. 2012) and Yu et al. (2015)). Even though injecting sulfur to one gridbox turned out to be an efficient injection strategy in the simulations of both models, the cooling is strongly concentrated over the tropics. The *Onegb* injection strategy might make the fundamental

problem of SRM, where tropics is cooled more at the expense of cooling of high latitudes, worse, if injections are concentrated in the tropics.

### 3.3.3   Sensitivity to injection altitude

Several studies have shown that the lifetime and the radiative forcing of stratospheric aerosols increase with the altitude of injections due to a longer sedimentation path (Heckendorn et al. (2009); Niemeier et al. (2011); Kleinschmitt et al. (2018);

Vattioni et al. (2019); Tilmes et al. (2018b)). Here we studied the impact of injection altitude using three scenarios, *Low*, *Baseline* and *High* scenarios, where sulfur is injected to 18-20 km, 20-22 km and 22-24 km altitudes, respectively. When compering scenario *High* to the *Baseline* scenario, our results with both models were consistent with the earlier studies. The injection rate did not have a large impact on how radiative forcing of sulfur injection to high altitudes compares with results of our *Baseline* scenario. Injecting sulfur to a higher altitude led to 14-16 % larger net radiative forcing compared to the *Baseline*

scenario when simulated with SALSA. With M7, scenario *High* led to 7-15% larger net radiative forcing than the *Baseline* scenario. As Fig. 7 shows, the injection to higher altitude led to effective radii values close to results of the *Baseline* injections while stratospheric sulfate burden was 12-20% larger in simulations with both models. This indicates that the larger radiative forcing in injection scenario *High* is caused mainly by a longer sedimentation path and the size distribution of aerosols is not significantly affected by the differences in the microphysical processes due to the injection altitude.

The impact of atmosphere dynamics on aerosol microphysics had a clearly larger role when injecting to lower altitude (18-20 instead of 20-22 km in *Baseline*). While the lifetime of aerosols was reduced as expected (Fig. 7a,b) because of the shorter sedimentation path, effective radii were also clearly smaller than in the *Baseline* scenario. This is consistent between the microphysical models. Simulations with SALSA showed that injecting sulfur to lower altitude enhanced net clear sky radiative forcing by 2-6% compared to radiative forcing in the *Baseline* scenario. In M7, the radiative forcing was 14-21% larger than in

the *Baseline* scenario in the case of injection rate of 2-20 Tg(S)yr−1, but 50 Tg(S)yr−1 injection rate led to roughly the same global mean radiative forcing as in the *Baseline* scenario.

Smaller aerosols in scenario *Low* compared to *Baseline* originated from differences in the atmospheric circulation at the different altitude. Fig. S7 in supplement shows the average meridional wind speed in the scenario *Low* with 5 Tg(S)yr−1 injection rates simulated with M7. In *Low* scenario, when injecting to lower altitude (18-20km), mean wind patterns point





from the Equator to higher latitudes. Winds are carrying more aerosols from Equator to high latitudes which reduces the sulfur concentration over the Equator compared to the *Baseline* scenario. This conclusion can be drawn also by analyzing where $SO_2$ is oxidized to sulfate. In the case of 5 Tg(S)yr−1 with M7 shows that 10-30% less sulfate is produced via $SO_2$+OH over the Equator in the *Low* scenario than in the *Baseline* while in the subtropics and mid latitudes sulfate production is 10-50% larger (supplement Fig. S7). As there is less $H_2SO_4$ gas over the Equator to condensate on the existing particles, particles are smaller

in *Low* scenario than in the *Baseline* scenario. Also due to the atmospheric circulation which carries more efficiently aerosols to higher latitudes, the zonal radiative forcing is concentrated more on midlatitudes than the tropics in case of injections to the lower altitude compared to the *Baseline* scenario Fig. 8.

  Our conclusions on the sensitivity of the radiative forcing to the injection altitude differ from the conclusions of Niemeier and Schmidt (2017). Their study showed that injecting sulfur to the altitude of 60 hPa (19km) resulted in a larger radiative

forcing than injecting sulfur to 30 hPa (25km), when the injection rates were larger than 10 Tg(S)$^{-1}$. Here in our simulations, scenario *Low* led to a larger radiative forcing compared to the injections to the altitude (20-22km) used in the *Baseline* scenario also in case of smaller than 10 Tg(S)$^{-1}$ injection rate. In addition, in the case of 50 Tg(S)$^{-1}$ injection rate simulated with M7, model results did not show a large difference in the global mean radiative forcing between scenarios *Low* and *Baseline*. When comparing to our scenario *High* (22-24km), which is close to higher altitude studied in Niemeier and Schmidt (2017),

the radiative forcing from *Low* injection strategy was higher only in the case of 5 Tg(S)$^{-1}$ injection rate simulated with M7. However in Niemeier and Schmidt (2017) sulfur were injected to one grid box while here sulfur were injected band over the longitudes and different model version and resolution was used. Overall this means that universal conclusion how injection altitude affects direct aerosol radiative forcing cannot be drawn as the results are depending on the atmospheric circulation in the altitude where injections take place, as well as injection rate and width of the injection area.

### 3.3.4 Sensitivity to temporal variation of injections

  In addition to scenarios where sulfur was injected continuously over a year, we studied two scenarios where the injections were concentrated only to a certain time of year. In scenario *Pulse2m*, injections were done during every second month starting from January and in *Pulse6m* injections were done in one month long periods twice per year, in January and July. Results from the *Pulse2m* scenario were close to results of the *Baseline* injection strategy in both models. One month frequency between

suspending injection and restarting it, is relatively short compared to the time required for transportation, oxidation or growth of the particles to have a significant impact on results compared to continuous injections over the year. Some of the aerosols moved out from the injection area during the pause in injections, which decreased condensation and coagulation on the existing particles over the Equator compared to the *Baseline* scenario. However, in the *Pulse2m* scenario injection rate at the time of the injection is doubled. In SALSA, the latter does not decrease radiative forcing efficiency as much as in M7 due to more

efficient nucleation at the expense of condensation. Thus, in SALSA simulations, the lifetime of aerosols was roughly 5 % and the global mean radiative forcing was 10-15% larger in *Pulse2m* than in the *Baseline* scenario. Simulation with M7 did not show a significant difference in the global mean radiative forcing between scenarios *Baseline* and *Pulse2m* (-3 - 7%).





In the *Pulse6m* scenario, the amount of injected SO$_2$ was 5 times larger during the injections compared to continuous injections. However, there was a 5 month period where the injections were suspended and aerosols had time to transfer to higher latitudes, before the next injection period was started. Thus there were less particles present at the injection are from the preceding injection where injected sulfur could condensate or where new nucleated particle could coagulate with. This reduces particles average size compared to *Baseline* injections. The zonal effective radius was smaller in all latitudes compared to the *Baseline* scenario in both models and all injection magnitudes (see Fig. 9). This enhanced the lifetime of aerosols by 17-35% with M7 and 20-27 % with SALSA. Based on SALSA simulations, the forcing efficiency of the *Pulse6m* scenario were largest of all studied scenarios as the forcing was over 30 % larger than in the *Baseline* scenario. The results of M7 show even a larger radiative forcing of the *Pulse6m* scenario compared to *Baseline* and its relative radiative forcing increased gradually from 40% to 58% when the injection rate was increased from 2 to 50 Tg(S)/yr.

It is expected that the results of the scenario, where sulfur is injected only during certain months, are sensitive to which months injections take place and how long the injection periods are. Atmospheric circulation varies during the year and the transportation and also the growth of aerosols are dependent on the timing of the injections. In addition, the seasonal cycle of solar radiation and how it coincides of the aerosol field has a large impact on the efficiency of SAI as well as the available OH for oxidation of SO$_2$. As shown by Visioni et al. (2019), radiative forcing is significantly dependent on the season, when injections takes place.

### 3.3.5 Seasonally changing injection area

Last of the studied scenarios was one where injections are varied seasonally, as suggested by Laakso et al. (2017). The aim of this type of strategy is to increase the efficiency of SAI compared to continuous Equatorial injections by targeting aerosol fields to coincide with maximum solar radiation during the year and reducing the particle size by varying the injection area so that sulfur is not always injected to regions where there are pre-existing larger aerosols from preceding injections. Another objective is to produce relatively less cooling in the tropics and more on mid and high latitudes compared to continuous Equatorial injections. This could prevent the over cooling of the tropics and under cooling of polar regions in the case of where average GHG-induced warming is offset by Equatorial injections (Laakso et al. (2017)).

Figure 6 shows that based on SALSA simulations, the global mean clear sky radiative forcing of scenario *Seasonal* was 10-20% larger compared to Equatorial injections. Simulations with M7 showed also a significant enhancement in the global mean radiative forcing especially with injection rates higher than 5 Tg(S)$^{-1}$ for which the radiative forcing was 45-57% larger than in the *Baseline* scenario. The size of the stratospheric aerosol was also smaller as Fig. 7 shows that the effective radius of stratospheric aerosols in the case of seasonal injections was clearly smaller compared to Equatorial injections. Further, Fig. 8 shows that the zonal mean radiative forcing was concentrated in the mid-latitudes rather than over the tropics.

The results of the *Seasonal* injection strategy was shown to be sensitive to the phase in which the injection regime was varied i.e if injection area has north-/south most position in boreal summer/winter Laakso et al. (2017). This affects the probability of optimal size particles coinciding with maximum incoming solar radiation, how fast SO$_2$ is oxidized, and also how aerosols are transported in the atmosphere because OH concentration and atmospheric circulation varies during the year. Different phases





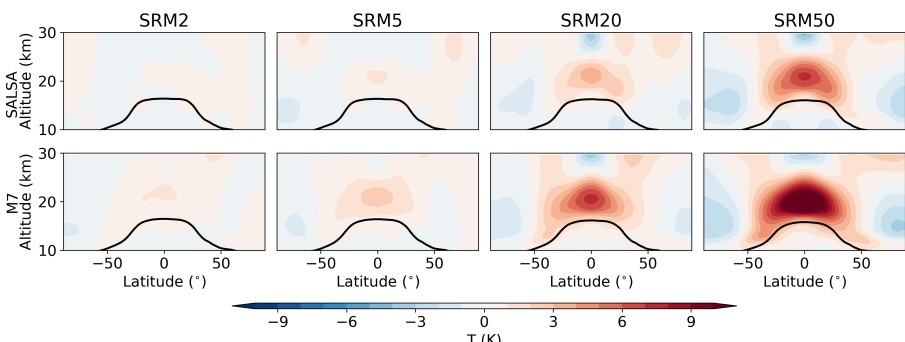

**Figure 10.** Temperature anomaly due to the *Baseline* stratospheric sulfur injection with different injection rates simulated with SALSA (upper panels) and M7 (lower panels). Black line indicates tropopause.

of seasonally varying injection are in the case of 5 Tg(S)yr$^{-1}$ was studied with ECHAM-HAMMOZ with SALSA in Laakso et al. (2017). In that study, none of the studied seasonally varying injection strategies led to a higher than 3% global mean radiative forcing compared to Equatorial injections, while in our current study the seasonal injection with 5 Tg(S)yr$^{-1}$ lead 580 to 10 % larger radiative forcing. However, there are several differences between the simulations here and simulations done in Laakso et al. (2017). The vertical resolution of in the model simulations in Laakso et al. (2017) was 47 levels which is not enough to reproduce the QBO (here 95 vertical levels were used). Injection strategies were slightly different e.g sulfur were injected at 20km of height (here 20-22km); in most cases injection regime varied between 30° N and 30° S (here 40° N and 40° S); here the northern-most position of the injection regime is reached in May, while in Laakso et al. (2017) studied scenarios 585 where north-most position were April and June. In addition, here a newer version of ECHAM-HAMMOZ is used.

### 3.4 Dynamical changes in the stratosphere and effects on the quasi-biennial oscillation

As previous sections has concentrated mainly aerosol microphysics and its impact on lifetime of aerosols radiative forcing here we study shortly changes in atmospheric dynamics. Stratospheric aerosol fields absorbs radiation which in turn warms the stratosphere. When sulfur was injected in the stratosphere, the warming it induced was strongest in the latitudes where the 590 aerosol fields were located (Fig. 10). Increasing the magnitude of injections led to stronger warming in the stratosphere and the temperature anomalies were significantly higher in M7 than SALSA as expected based on the amount absorbed LW radiation (Fig. 1). Based on SALSA, warming anomalies inside the injection regime (10° N - 10° S, 20-22 km altitude) were 0.39, 1.07, 3.26 and 6.83 K for 2, 5, 20 and 50 Tg(S)yr$^{-1}$ injection rates respectively while the corresponding temperature anomalies were 1.04, 2.28, 6.84 and 11.48 K in M7 simulations.

595 Stratospheric warming was concentrated in the tropics in all studied SAI scenarios, but the magnitude of warming depended on the injection strategy (see supplement Fig. S8-9). Injecting sulfur to a narrow band over the Equator (*Narrow*-scenario) led to a stronger stratospheric warming than seen in the *Baseline* scenario, while in scenario *Wide* there was less warming. As expected, varying the injection area seasonally (Seasonal) did not warm stratosphere as much as the other scenarios.





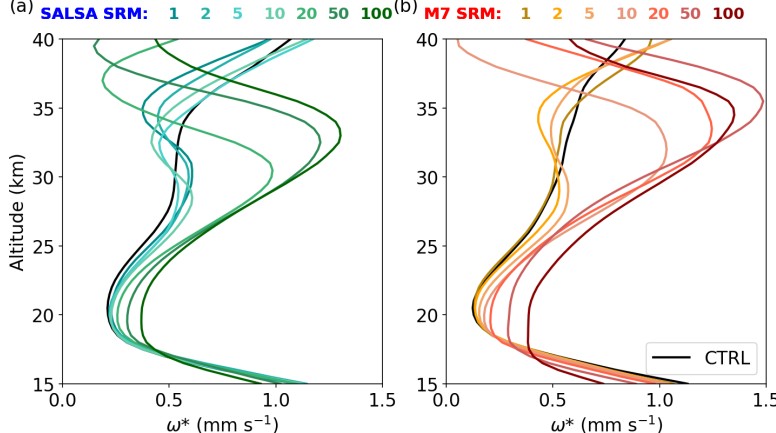

**Figure 11.** Residual vertical velocity in the tropics ($10° N - 10° S$ latitudes) for different injection rates in simulations with a) SALSA and b) M7

As the tropical stratosphere warms, it changes the dynamics of the atmosphere: for instance, it leads to a stronger vertical advection which further strengthens the lofting of aerosols and makes the lifetime of aerosols longer. Figure 11 shows the residual vertical velocity ($\omega^*$) between $10° N - 10° S$ latitudes in baseline scenarios with the different injection rates. Higher injection rate causes stronger warming which further strengthens $\omega^*$ at the injection altitude (20-22 km) and even up to 30 km altitude. As expected, based on the stronger warming seen in M7 than SALSA (Fig. 10), the increase in the updraft velocity is generally higher in M7. In case of the 100 Tg(S)yr$^{-1}$ in injection rate in M7, at the 20km altitude the increase in $\omega^*$ was 360 %. However, note that the profiles of $\omega^*$ in the CTRL simulation without SAI were significantly different between SALSA and M7. This is discussed later in this section. One interesting feature is seen in altitudes higher than 25 km, where the $\omega^*$ changes drastically when the injection rate is large enough. In SALSA, this takes place in injection scenarios higher than 10 Tg(S)yr$^{-1}$ while in M7 this happens already with injections higher than 5 Tg(S)yr$^{-1}$. While in lower than above-mentioned injection rate, $\omega^*$ is roughly 0.5 mms$^{-1}$ at the 30km altitude, in case of larger injection rate $\omega^*$ is larger than 1.0 mms$^{-1}$. Tropical $\omega^*$ in different injection scenarios compared to baseline scenario with injection rates 5 and 50 Tg(S)yr$^{-1}$ are shown in supplement Fig. S10.

Changes in zonal and meridional wind patterns in December-January-February and June-July-August are shown in the supplement (Fig. S11-14). Zonal wind increase in the tropics is stronger with higher injection rates and in M7 than in SALSA. The meridional wind pattern anomalies in SAI scenarios have in some cases different signs between SALSA and M7. Stratospheric sulfur injections have also been shown to e.g strengthen the stratospheric polar vortex Visioni et al. (2020a). It is expected that these changes are also sensitive to the aerosol model as well as to the injection strategies. However, a more detailed analysis of this is beyond of the scope of this study.

One consequence of the warming of the tropical stratosphere is the slowing down of the quasi-biennial oscillation (QBO) and if the injection rate is high enough, the QBO can be shut down completely ((Niemeier and Schmidt, 2017)). This statement



was supported also by our simulations. As can be seen in Fig. 12, the QBO was slowing down with increasing injection rates. Based on M7 simulations, the shutting down of QBO takes place for injection rates higher than 10 Tg(S)/yr while in SALSA more than 20 Tg(S)yr$^{-1}$ is required. In addition, after the shutting down of the QBO, the westerly phase of QBO in the lower stratosphere is stronger and it reaches higher altitudes in M7 compared to results of the corresponding injection magnitude in SALSA. In M7, stratospheric heating due to stratospheric aerosols was stronger than in SALSA and thus the QBO slows down

and vanishes with lower injection rates. A similar difference in the impacts on the QBO between climate models has been seen in Niemeier et al. (2020) which were caused difference in residual vertical velocity between models and different heating rates in the lower stratosphere. Probably also here, difference in responses in QBO between models are not fully caused by aerosol microphysics.

As Fig. 11 and Fig. 12 show, residual vertical velocity and QBO were different between the aerosol schemes initially even

without the stratospheric sulfur injections, even though the same atmospheric model was used. In the CTRL simulations, M7 has a much longer period of QBO than in SALSA and it is overestimated compared to 28 months seen in observations (Naujokat (1986)). Even though the same atmospheric model was used, different tuning parameters are used depending on the aerosol microphysical model and the atmospheric temperature is not consistent between models even in CTRL simulation (supplement Fig. S15). Based on test simulations, which were performed before the actual simulations of this study, the tuning of the

model had a significant impact on the QBO. In addition, even though aerosol concentration in the stratosphere is low in CTRL simulations, the tropospheric aerosols and the following indirect cloud impacts are different between the aerosol schemes used. In M7 simulations, modes were modified to represent stratospheric aerosols more accurately by narrowing the standard deviation of the coarse mode and changing the threshold radius when aerosols from the accumulation mode are transferred to the coarse mode. A disadvantage in this modified setup is that large tropospheric aerosols are probably not represented as

well as in the standard setup. Differences in tropospheric aerosols and tuning parameter led to warmer upper troposphere and lower stratosphere in SALSA than in M7 as seen in supplement Fig. S15. Thus residual vertical wind (Fig. 11), zonal and meridional wind patterns and QBO in CTRL simulations are different between SALSA and M7. This also will lead some bias in atmospheric circulation in the studied SAI scenarios between the models.

Franke et al. (2021) and Kravitz et al. (2019) have shown that the response of the QBO to sulfur injections depends on the

location of injections. In Franke et al. (2021) simulation where sulfur was injected to two gridboxes located at 30° N and 30° S and in Kravitz et al. (2019) were sulfur was injected to 4 different latitudes based on a feedback algorithm, the QBO was not significantly affected even with 25 Tg(S)$^{-1}$ injections. QBO in our sensitivity scenarios is shown in the supplement (Fig. S13). Studied scenarios here included injections at the Equator. However, in scenario Seasonal, the injection region varied and sulfur was injected to Equator only during a part of the year. In this case, sulfur injection did not have a significant impact

on QBO even if 50 Tg(S)$^{-1}$ were injected in SALSA simulation. The results of the corresponding scenario with M7 showed that QBO was prolonged with 20 Tg(S)$^{-1}$ and shut down with 50 Tg(S)$^{-1}$ injection rates. In all the other scenarios, the QBO vanished with 50 Tg(S)$^{-1}$ injection rate and in most scenarios with an injection rate of 20 Tg(S)$^{-1}$ in both models. In *Low*, *High* and *Wide* scenarios QBO did not fully disappear in the case of 20 Tg(S)$^{-1}$ injection rate with SALSA, but the cycle were significantly prolonged.





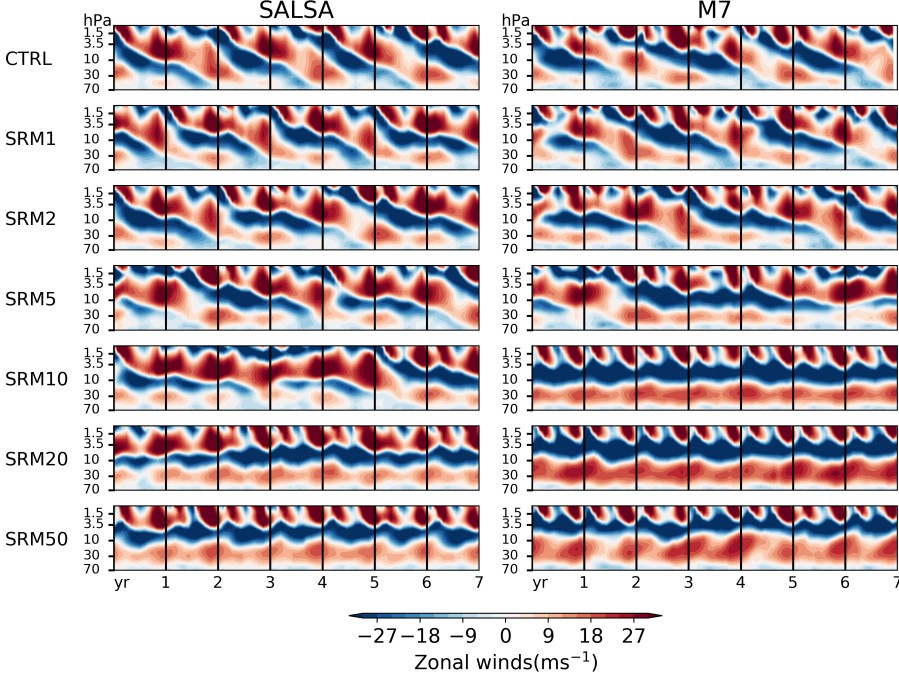

**Figure 12.** Zonal mean zonal wind (ms-1) at the Equator for CTRL and stratospheric sulfur injection with different injection rates. SALSA results are shown in left panels and M7 results are in right panels

## 4   Conclusions and discussion

Here we have systematically studied different spatio-temporal injection strategies with different magnitudes (2-100 Tg(S)$^{-1}$) of sulfur injections using both the sectional (SALSA) and modal (M7) aerosol schemes in ECHAM-HAMMOZ. These simulations showed significant differences in SW and LW radiative forcings, size of the aerosols and lifetime of sulfur between the different injection strategies. In addition, modelled results were very sensitive to which microphysics model was used in the simulations. While both models show sublinear increase of the global mean net radiative forcing as a function of the amount of injected sulfur due to the increases in the size of aerosols, the net radiative forcing of SAI was 88-154% higher based on simulation with SALSA than M7. This large difference was also present when SW and LW radiative responses between models were compared individually. While SW radiative forcing was 45-85% higher (more negative) in SALSA than M7 with corresponding injection rate, LW radiative forcing was 33-67% larger in M7.

We identified two main factors which were causing different responses between models: 1) The numerical methods for describing the competitive processes of new particle formation and condensation inside the injection regime and 2) limiting the evolution and the shape of the particle size distribution due to mode setup in M7. In the stratosphere, new particle formation by nucleation is fed by continuous injections. On the other hand, there are already pre-existing particles from the preceding injections to which injected sulfur can condense. In SALSA, the nucleation rates are higher than in M7 and sulfuric acid gas





tends to form new particles rather than condensing on the existing ones. In M7 simulations the opposite is true, and sulfate tends to condense on pre-existing particles and leave less sulfur available for nucleation. Simulations with M7 showed that continuous injections and condensation increased the size of the largest (coarse) mode. However due to the setup for the repartition of particles between modes, this did not allow the second largest mode to grow, creating a gap in the particle size distribution between the two largest modes. This gap coincided with the particle size range which would be optimal to scatter

radiation.

Overall, differences in the results between the two microphysical models reveal significant uncertainties related to stratospheric sulfur injections. Thus, there is a need for better tools to analyze aerosol microphysical processes in stratospheric conditions under continuous injections and to improve the aerosol-climate models. A comparison with the observations of large volcanic eruptions (e.g Pinatubo) does not necessarily offer a true picture of the model performance under continuous

injections as there is not as large competition in nucleation and condensation in the case of volcanic eruptions, where a large amount of sulfate erupts abruptly to a relatively particle free stratosphere. It is also good to keep in mind that sulfur would most plausibly be injected to the stratosphere by aircraft (Smith and Wagner (2018)). In these simulations with the climate model the injected $SO_2$ is instantly mixed in the model grid box with the size of few hundreds square kilometers. Thus microphysics which is taking place inside of plume after injection is not captured by ECHAM-HAMMOZ and aerosol-climate models more

in general. In addition, the time step of climate models might be too long for aerosol microphysical processes which can cause strong bias for example here, were sulfuric acid concentrations are high and new particle formation and condensation is resolved simultaneously.

Uncertainties in microphysics will further lead to large uncertainties in estimating possible climate impacts of stratospheric sulfur injection or e.g for estimations of how much sulfur is required to achieve a certain climate target. To offer some perspec-

tive: compensating all radiative imbalance in the Representative Concentration Pathway (RCP) 2.6 at the end of this century ($2.6 \, \mathrm{Wm^{-2}}$) by stratospheric sulfur injections would require roughly yearly 3.7 Tg(S) injection based on the results of SALSA while based on M7, 15 Tg(S)yr$^{-1}$ is required to achieve this. The difference is even more significant in extreme cases where the radiation imbalance in the RCP 8.5 scenario ($8.5 \mathrm{Wm^{-2}}$) would be compensated by SAI. There, the estimation of required sulfur are either 22 Tg(S)yr$^{-1}$ or over 100 Tg(S)yr$^{-1}$ depending on the microphysical model. These uncertainties between

models have also a significant impact on uncertainties related to the global mean precipitation. Larger LW absorption in M7 compared to SALSA might translate to a significant reduction of the global mean precipitation which has been shown to correlate negatively with absorbed radiation Laakso et al. (2020). Lower net radiative forcing in M7 means that more sulfur should be injected to get the same cooling impact as in SALSA, which means even stronger absorption of LW radiation and decrease in the global precipitation. This will be studied further in part 2.

We also simulated different stratospheric sulfur injection strategies with both microphysical models. These scenarios were simulated with 2, 5, 20 and 50 Tg(S)yr$^{-1}$ injection rates. We studied how choices of injecting to narrow vs. wide latitude bands, high vs. low altitude, to one gridbox vs. band over longitudes and temporal differences between injection strategies (pulsed and seasonally changed) affect the radiative forcing of SAI and if the results are consistent between the models.



Differences in all-sky radiative forcing of the most efficient injection strategy compared to the least efficient strategy were
33-42% higher depending on the injection rate based on SALSA. Simulations with M7 showed even larger variation in radiative
forcing and all sky radiative forcing in the most efficient SAI scenario was 48-216% higher than in the least efficient simulated
scenario. However if we exclude the *Onegb* scenario with 50 Tg(S)$^{-1}$ injection rate, which showed spurious results, probably
due to numerical issues, the most efficient SAI scenario was 76% larger than in the least efficient scenario based on M7
simulations.

Because the forcing efficiency decreases with the injection rate, the injection strategy would matter especially in the case
of high injection rate. As an example, based on M7 simulations, in case of 20 Tg(S)$^{-1}$ injection rate, using seasonal injection
strategy instead of using Equatorial injections (*Baseline*) would enhance the global mean all-sky radiative forcing by 53%. To
achieve the same enhancement using Equatorial injections would require 42 Tg(S)$^{-1}$ injection rate instead of 20 Tg(S)$^{-1}$.

Our *Baseline* scenario, where sulfur was injected continuously between $10°$ N and $10°$ S latitudes at 20-22 km altitude
resulted in, depending on the injection rate, the smallest or the second smallest net radiative forcings of all studied injection
strategies. Only the injection strategy *Narrow* (injecting in a two grid boxes-wide band over the Equator) led to a smaller
radiative forcing when simulated with M7 and in the case of 5 and 20 Tg(S)yr$^{-1}$ injection rates. In SALSA, the *Wide* injection
scenario was the least efficient when less than 20 Tg(S)yr$^{-1}$ was injected. Based on the simulations, injecting sulfur to lower
altitudes (18-20 km) was more efficient than injecting sulfur to 20-22 km altitude in both models with the exception of the
simulation with M7 for 50 Tg(S)yr$^{-1}$ injection rate.

If the studied injection strategies are ranked based on their global mean radiative forcing, there are not large differences in the
order whether SALSA or M7 results were used. Injecting twice per year, in one-month periods, was the most efficient injection
strategy based on SALSA and and M7 with the exception of 50 Tg(S)yr$^{-1}$ when simulated with M7, where the continuous
injection to one grid box resulted in a suspiciously large impact. Generally, relative differences in the global mean radiative
forcings between different the injection strategies were larger when simulated with M7 than SALSA. Overall results from both
models indicated that injection over the area where large aerosols from preceding injections already exists would lead to higher
condensation on the existing particles, or that new particles will coagulate with the existing ones, which reduces the efficiency
of geoengineering.

Zonal mean radiative forcings dependence on the injection strategy were qualitatively similar between the models. If com-
paring with the *Baseline* scenario, injecting sulfur to a narrow band or to one model grid box increased the radiative forcing
over the tropics, as expected. On the other hand, injecting sulfur to a lower altitude, wider band over the Equator or changing
injection area seasonally led to reduced radiative forcing over tropics compared to our *Baseline* scenario. For example, com-
pared to the *Baseline* scenarios, changing injection area seasonally led in SALSA to 29% and in M7 75% larger all sky radiative
forcing over non-tropics in case of the injection rate of 5 Tg(S)yr$^{-1}$. In the Tropics, corresponding changes were 13% smaller
and 12% larger with SALSA and M7 respectively. Thus, seasonally varying sulfur injections might prevent overcooling of the
tropics and warming at high latitudes. This has been seen in several studies where GHG-warming has been offset with SRM.
This could be done without any trade off in the total radiative forcing, as net all sky radiative forcing of seasonally changing
injection strategy were one of the most efficient of the studied scenarios. Since it also reduced LW absorption compared to the





*Baseline* scenario, it would thus lead to smaller reduction in global mean precipitation (Laakso et al. (2020)). In addition, based

on SALSA simulations, the *Seasonal* scenario did not have a significant impact on the slowing down or complete disappearance of the QBO even with 50 Tg(S)/yr injections.

Here the simulations were done with a model configuration with fixed SST and the object was to analyze the impact of injection strategies and aerosol microphysical models on radiative forcing. The next step is to study how these radiative forcings translate to climate impacts, i.e changes in global and regional temperature and especially precipitation and if these responses

are dependent on the climate model. This will be studied in part 2.

*Data availability.* The data from the model simulations and implemented model codes are available from the authors upon request.

**Appendix A: Simulations without nucleation**

Our analyses in this study showed that model responses on stratospheric aerosol injection were significantly different between sectional aerosol model SALSA and modal aerosol scheme M7. Based on Fig. 4, this difference between the models is

partly caused by how models solve the competition between new particle formation and condensation inside or near the injection regime. While injected sulfur tends to mainly form new particles in SALSA, in M7 sulfate condensates on pre-existing particles. To study the role of these differences on radiative forcing and particle size distribution we performed additional simulations where competition between nucleation and condensation were made consistent between the models. This was done by switching nucleation off, and emitting 25% of injected sulfur as $3\,\mathrm{nm}$ primary particles while the rest of the sulfur was injected

as $SO_2$. Simulations were done for *Baseline* scenario with 2, 5, 20 and 50 $\mathrm{Tg(S)yr^{-1}}$ injections. By standardizing the competition between nucleation and condensation in SALSA and M7, the global mean SW radiative forcing, net radiative forcing and especially stratospheric sulfur burden between models are closer to each other than in the original setup (see Fig. A1). However, differences in the LW radiative forcings between models were slightly increased and the SW response was still significantly larger in SALSA than M7. Thus, the different treatment of the competition between nucleation and condensation explains the

responses between models only partly. This is interesting especially because the zonal mean effective radius of stratospheric aerosols is consistent in these sensitivity simulation between models (See Fig. A2). However, a closer look on the aerosol size distribution inside the injection regime shows that even the effective radius were consistent between models, the size distribution is not. Generally, size distributions in simulations where nucleation is replaced by injecting 3nm primary aerosols are relatively similar to those in the original simulations, excluding the two smallest modes in M7 simulations (Fig. A3). The me-

dian size of nucleation (smallest) mode is larger, the number of Aitken mode (second smallest) aerosols is significantly higher and the median size is smaller than in the original simulations. Because of this, the effective radius of stratospheric aerosols is significantly smaller. However the aerosols in nucleation and Aitken mode do not have a notable impact on the radiation. As Fig. A3 shows, the two largest modes, accumulation and coarse, are relatively similar to those in the original simulations. There is still a large gap between these two modes, which is located at the size range of the largest backscattering efficiency. In



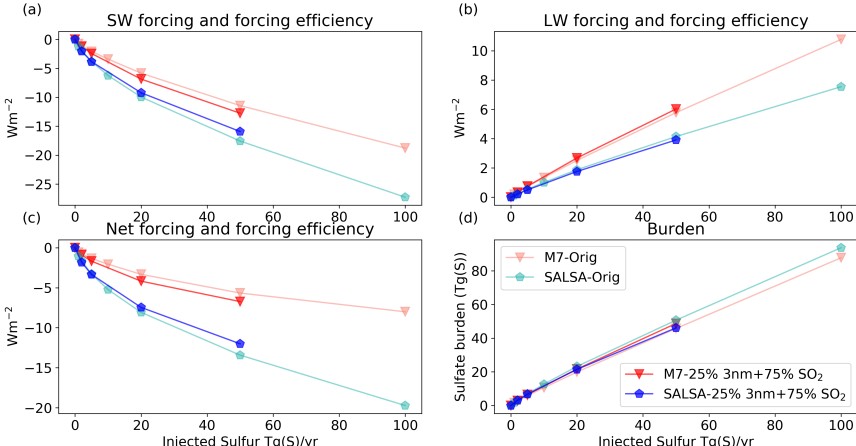

**Figure A1.** Global mean a) shortwave, b) longwave and c) total forcing and d) stratospheric sulfate burden as a function of injected sulfur. M7 results are shown by red lines and markers and SALSA results are indicated by blue color. Lighter dashed lines correspond original simulations and darker solid lines are representing corresponding simulations without the nucleation.

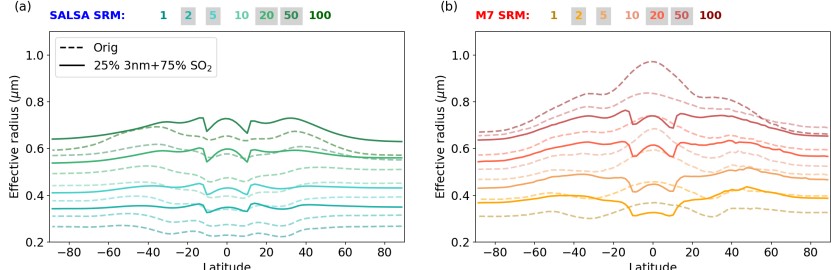

**Figure A2.** Zonal mean effective radius of stratospheric aerosols in different magnitude of sulfur injections simulated with a)SALSA and b) M7. Lighter dashed lines correspond original simulations and darker solid lines are representing corresponding simulations without the nucleation.

addition, the number of coarse mode aerosols is even higher than with the original setup which explains the larger LW radiative forcing.

Overall, these simulations show that different radiative responses between the models are mainly originated from the representation of aerosols (sectional vs. modal), which were discussed in sect.3.2. In addition, this analysis shows that the effective radius is not always a representative measure for the size distribution and radiative impacts can be significantly different even

in case of consistent effective radius.





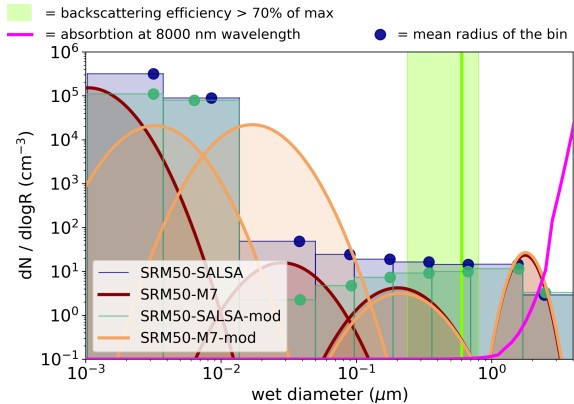

**Figure A3.** Aerosol number size distribution inside of injection regime in scenarios SRM5 and SRM50 simulated with M7 and SALSA without the nucleation. Dots on the top of the SALSA size bins are showing a mean diameter of the bin. Light green line shows to size of the maximum back scattering and shaded area indicates radius where aerosol backscattering is 70 % of maximum Dykema et al. (2016). Magenta line shows dependence of absorption of 8000nm wavelength on the (dry) size of the sulfate aerosols, based on the radiation module of SALSA.

*Author contributions.* AL designed the research, performed the experiments, carried out the analysis, and prepared the paper. All authors contributed ideas, participated in interpretation and discussion of the results, and contributed to writing the paper.

*Competing interests.* The authors declare that they have no conflict of interest.

*Acknowledgements.* The ECHAM-HAMMOZ model is developed by a consortium composed of ETHZ, Max-Planck Institut für Meteorolo-
gie, Forschungszentrum Jülich, the University of Oxford, and the Finnish Meteorological Institute; it is managed by the Center of Climate Systems Modeling (C2SM) at ETHZ.

*Financial support.* This research has been supported by the Tiina and Antti Herlin Foundation (grant no. 20190014 and 20200003).





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
