# Peer review of "Dependency of the impacts of geoengineering on the stratospheric sulfur injection strategy part 1: Intercomparison of modal and sectional aerosol module"

_Atmospheric Chemistry and Physics, 2021_

## Author Response (AR1)

We thank Ben Kravitz for suggestions and comments. Comments helped to clarify several parts of the text. Our point by point answers to the comments are presented below. Referee comments are in bold and our replies in body text.

**This is a great study, and important. This has needed attention for some time. I am recommending minor revisions. I'd like to see the conclusions fleshed out a bit more. What have we learned about best practices for simulation? When is it okay to use a modal model versus a sectional one? I'd like to see some insight that people in the field can use.**

Thank you for this comment. We added some discussion on best practices in the text. However we also want to emphasize that we purposely avoided making too strong suggestions to use either one of the models used here or sectional or model models generally. The model capability to simulate stratospheric aerosols depends on several different factors e.g how individual microphysical processes are represented in the model in addition to how aerosol size distribution is represented. Even though we strove to analyze those comprehensively, there is the need for more extensive research for this topic which covers multiple models. In any case we now focus more on practices and issues which should be considered when analyzing results of modal/sectional models or when choosing between those (includes suggestions from the other reviewer).

We changed the last line of section 2.1: "*However when using M7, this requires changes in the configuration of the modes and having a narrower width of the largest mode to improve the representation of the stratospheric aerosols (Kokkola et al., 2009). Thus one downside of using a modal scheme is that tropospheric and stratospheric aerosols are not well described with the same mode configuration*"

We also added the following paragraph to Conclusions:
"*There are several factors which support a sectional model over a modal model for stratospheric aerosol simulations, despite the fact that the modal scheme is significantly computational faster than sectional (simulations with M7 were 60% faster than SALSA). First of all, tropospheric and stratospheric aerosols require different configurations for modes and thus studying both in the same simulations is not recommended. In addition, even though only stratospheric aerosols are studied, the tropospheric aerosols, which were not well represented by configuration designed for stratospheric aerosols, can affect indirectly to stratospheric aerosols. In SAI simulations, and especially in the case of continuous injections, the size distribution inside the injection region does not have a clear multimodal structure in the sectional model simulations except for the lowest injection rates (1-2 Tg(S)yr-1) (Fig 5 and Fig A3 ) (English et al, 2012, Kleinschmitt et al., 2018). This is probably because there is available H2SO4 gas for particles to grow by condensation, and particles are not accumulating to certain size classes by coagulation. This kind of size distribution cannot be represented by 4 modes and in this study the problem culminates in that there is a persistent gap between the two largest modes. One option could be to use more modes, but then the computational benefits compared to sectional schemes would become smaller. In the standard setup of M7 (the largest) coarse mode width is 2.0 instead of 1.2 which is used here. This would make the gap between the two largest modes smaller. However in the case of stratospheric sulfur injections or a large volcanic eruption, a wider coarse mode width leads to a tail of large particles. This causes an overestimation of the*

*effective radius of the coarse mode and increased the sedimentation velocity and reduced residence time of aerosols in the stratosphere which is the reason why the different setup is used for stratospheric aerosols. One option could be to increase mode widths of the Aitken and accumulation modes. However, number concentrations of these modes are typically higher and thus widening of the modes can lead to a situation, where widened mode would cover the adjacent larger mode. It is also good to keep in mind that the partitioning of sulfuric acid to particle phase due to nucleation over condensation was suspiciously large in SALSA and the model produced significantly larger net total radiative forcing than in e.g. Kleinschmitt et, al. (2018), where simulations were done with the sectional model. Thus, even though there was not as clear a shortcoming as the gap between modes in M7, there is a need to analyze the individual microphysical processes and to understand the differences between the results of different sectional models.*"

**I found numerous typos, LaTeX issues, and other errors at the word and sentence level, both in the text and figures. Another round of proofreading (or having ACP do copyediting) would be useful.**
We did a round of proofreading and tried to fix issues.

**I found the description on lines 241-248 confusing. You first attribute the difference between the studies to different LW treatments, then the optical properties, and then the different size distributions. Which is it and how do you know?**
We added the word "probably" to line 243, which now reads:
"*This indicates that differences in radiative forcings between the studies are probably caused by differences in the LW radiation transfer, i.e in using a different radiative transfer scheme, or differences in the aerosol optical properties in LW radiation calculations.*"
We just wanted to bring up possible explanations for different results between our study and Kleinschmitt et al. (2018). To say which one is causing the differences, or if there is some other explanation, we would need to have more information about Kleinschmitt et al. (2018) results and about the model used in their study.

**Line 267: I see four modes in Figure 2, so it's not obvious to me which one you're calling the accumulation mode. I'm guessing the second-largest one, since that would fit with small changes in number concentration?**
The reviewer is correct. It is the second largest mode. We added the name of the modes in the caption of Figure 2, which now reads:
"*Figure 2. Aerosol number size distribution at the Equator and at 20-22 km altitude in scenarios with different injection rates simulated with M7 (names of the four modes from the left: Nucleation, Aitken, Accumulation, Coarse) and SALSA (10 size bins, there is no significant number of aerosols in largest bin, and thus it is not shown in the figure.). Dots on ...*"

Note that we also changed Figure 2 based on comments from the other reviewer.

First line in section now reads (2.1.2): "*In M7, aerosols are represented using a superposition of seven log-normal modes, 4 for soluble (nucleation, Aitken, accumulation and coarse) and 3 (Aitken, accumulation and coarse) for insoluble material.*"

**Line 268: How does the coarse mode change in size? That capability/feature seems like it should be mentioned in Section 2 in your description of M7.**

We modified section 2.1.2 which now reads:

*"Similarly to Niemeier and Timmreck (2015), we modify the setup of the modes so that the coarse mode is made narrower than in the standard setup (using standard deviation $\sigma_{CS}$ = 1.2 instead of 2.0). In the case of high sulfur concentrations, a coarse mode with 2.0 standard deviation has been shown to lead to a tail of large particles (Kokkola et al., 2009). Based on Kokkola et al. (2009) this caused an overestimation of the effective radius of the coarse mode, when compared to the highly resolved particle spectrum reference model, and thus increased the sedimentation velocity and reduced the residence time of aerosols. In M7 the median size of the mode can change, but only between mode specific maximum and minimum threshold radii. For the nucleation mode there is no lower threshold radius and for coarse mode there is no higher threshold radius. This threshold radius also defines when aerosols are transferred from one mode to another. As in Niemeier and Timmreck (2015) we changed this threshold radius between accumulation and coarse mode (the two largest modes) from 0.5 μm to 0.2 μm."*

**Line 270: You could quantify "considerably different" by integrating the area under the pink curve that falls within the coarse mode.**

Unfortunately, this line was written confusingly. Line 270 was not meant to refer to the previous line. Here we wanted to say that generally speaking, the size distributions between our study and Niemeier and Timmreck 2015 are considerably different.

Line 270 now starts the new paragraph and now reads: *"Compared to the aerosol size distribution in Niemeier and Timmreck (2015), the size distribution based on M7 simulations in our study was considerably different."*

**Lines 275-276: As written, this is kind of a throwaway claim. You have to show that this matters if you're going to claim it. Same comment for lines 467-468 and line 500.**

Lines 275-276 is rewritten as following: *"In addition, the difference between our study and Niemeier and Timmreck (2015) is a different version of the GCM and different resolution used in the model. Niemeier and Schmidt (2017) have shown that low and high vertical resolutions led to different stratospheric dynamics which further caused differences in aerosol sizes in SAI simulations"*.

We are now citing Niemeier and Schmidt (2017) on lines 467-468.

We added *"Our results indicate that "* to Line 500 as this claim is discussed in the text and is based on the results: *"Our results indicate that the impact of atmosphere dynamics on aerosol microphysics had a clearly larger role when injecting.. "*.

**Lines 291-297 (and also section 3.2): So which one is right? My guess would be that SALSA is closer to the truth (although certainly not perfect), and I acknowledge that this is hard to verify based only on Pinatubo, but I'd like to see you talk more about this.**

We do not want to make any strong statements which one is right. As you mention, it is really difficult to verify. Obviously there are some numerical issues especially in M7 and the

gap between the modes which we wanted to bring up. Our main message in this study is that there are a lot of uncertainties when modelling SAI and there is a certain need to decrease these uncertainties and improve our aerosol models. Anyway now we discuss this more in conclusions (see our reply for first comment).

**Lines 618ff:  True, for equatorial injection (as you say later).**
This is now rewritten as: "*One consequence of the warming of the tropical stratosphere in case of equatorial injection is the slowing down of the quasi-biennial oscillation..*"

Other corrections:
1) "and forcing efficiency" is removed from Figure A1 panel titles
2) Caption of Fig A3 is fixed, colors are changed and number distribution of M7 simulations is now shown as sum of the modes (thus not showing individual modes separately).
3) We added a comment to Section 2.2 that concentration of injected sulfur varies in the seasonal scenario during the year: *"Note that as injection band is always 20° latitudes wide and a same mass in injected in every month, the concentration of injected sulfur is smaller during the times when the injection area is located over the Equator compared to times, when it is over midlatitudes"*

We thank Peter Irvine for his thorough review, great suggestions and comments which improved the manuscript. Our point by point answers to the comments are presented below. Referee comments are in bold and our replies in regular text. Note that names of three of the simulated scenarios have been changed, but we still use original scenario names here in our replies so that they correspond to scenario names in reviewer comments. (*r1) means that we are referring to this reply later.

**The authors use a state-of-the-art climate model coupled to two different aerosol modules to simulate a wide range of stratospheric aerosol injection (SAI) deployment strategies and deployment scales. The authors find that the sectional model produces a substantially larger negative shortwave forcing and smaller positive longwave forcing, and hence an even greater net radiative forcing, than the modal model. The main driver of this difference is the competition for gaseous sulfate between the nucleation of new particles and condensation onto existing large particles, with more of the sulfate captured by large particles in the modal model. The authors note that due to limitations in the microphysical representation in the modal model, a "modal gap" (my term) opens up between efficiently-scattering accumulation mode particles and coarse mode particles that falls within the range where particles most effectively scatter light, producing a drop in SW forcing. The authors find that in both models the net forcing from SAI increases sub-linearly with deployment scale, with a greater decline in forcing efficiency in the modal model. Almost all alternative deployment scenarios produce a greater forcing than the baseline scenario (10N-10S, all longitudes). Scenarios which inject the aerosols over a wider range of latitudes or at different latitudes in different seasons avoid an over-concentration of particles in the tropics which promotes the growth of larger, shorter-lived particles in the baseline scenario. Both the high and low altitude deployment increase forcing, for the high scenario this is a result of increased lifetime, and for the low scenario this is due to increased polewards dispersion. Pulsed scenarios also increase the total forcing by increasing the concentration of gaseous sulfate to existing particles, promoting nucleation and the growth of smaller particles.**

**This paper makes a novel and important contribution to the literature on SAI, making the most thorough evaluation of SAI deployment scenario in a state-of-the-art model and doing so with two different aerosol modules. It is generally well-written, presented and argued, and I learned a lot reading it. However, I feel the authors need to do more to communicate the implications of these findings for the field.**

**For the scenario choice, this could be addressed straightforwardly with a more pointed discussion of the usefulness of focusing on equatorial injections as the baseline case for evaluating SAI. This paper piles yet more evidence on to the case that they are inefficient and produce an uneven, tropics-heavy sulphate burden, shortcomings that can be readily avoided with alternative deployment choices. GeoMIP G6sulfur was an equatorial strategy, what do these results suggest about the wisdom of the next generation of GeoMIP focusing on equatorial injection?**

This is a good point. As the problem is complex we do not want to give any direct general suggestion for an alternative baseline case. We now discuss this in the conclusions section:
"*Most of the SAI studies have been focusing on equatorial injections as the baseline case. Injection between 10N to 10S was also chosen the injection scenario in G6sulfur GeoMIP*

*experiment (Kravitz et al., 2015). As results here show, there are several factors than indicate that equatorial injections should not be used as baseline scenario: 1) It is less efficient than most of the studied alternative injection scenarios, 2) the resulting radiative forcing is concentrated in the tropics and 3) the warming of tropical stratosphere leads to a slowing down or vanished QBO, which could be avoided by some other injection strategies such as varying the injection area seasonally. However it is not straightforward to give any suggestion for alternative baseline scenario for following studies or e.g next possible GeoMIP experiments, as SAI can be used to meet various different climate targets and none of the injection strategies can be optimized to meet all of the targets. In addition, there were e.g some changes in the mutual ranking in global mean forcing between studied scenarios, depending on injection rate. This is true especially when considering injection rates which can be considered "more realistic" (<10 Tg(S)/yr). In scenarios like G6sulfur, where injection rate varies, the most logical scenario for lower injection rates may not be the same as for higher injection rates. As this study shows, the radiative forcing of certain injection strategies is also significantly dependent on the aerosol model. Thus there is a need for model intercomparison project using aerosol-climate models to simulate various SAI scenarios. "* (*r1)

**While I'm not an expert on aerosol microphysics, it seems to me that the authors present evidence (discussed around line 360) that suggests modal models can, or at least the modal model they employ did, produce an unrealistic simulation of SAI for large deployments. Later, the authors speak loosely of uncertainties when describing the differences between these two aerosol modules but what I'd like to see instead is much more expert judgment and discussion on the realism of the simulated aerosol distributions. It sounds to me that the "modal gap" that opens up, right around where aerosols most effectively scatter light, between accumulation-mode and coarse-mode aerosols is unphysical and represents a serious shortcoming in this representation of the aerosol microphysics of SAI.**

**The question of whether the simulated aerosol distribution is realistic in this modal model is crucial and the introduction and discussion do not do enough to set it up and answer it, with the main discussion of this issue buried in the midst of the results section. I'd like to see a paragraph or two devoted to the differences between modal and sectional representations of aerosol microphysics in the introduction that raises common shortcomings, challenges (presumably the possibility of a modal gap opening up has been discussed previously?) and the relative performance or realism of the different options. Similarly, I'd like to see a much more in-depth discussion of these issues in the discussion and conclusion. Would this issue have occurred in a modal model with more modes? Similarly, the discussion should address the question: "are modal representations appropriate for SAI given our findings?" and provide some guidance on their limits or warnings about particular setups to other modelers working on this topic so that they can represent SAI more realistically.**

We reply to two previous comments simultaneously. To our knowledge there are no earlier discussions about the model gap. Generally simulating aerosol microphysics in stratospheric conditions and possible shortcoming has not received much attention. Thus this topic is not much discussed in the introduction section but we now highlighted it more in the conclusion/discussion section.

Kokkola et al (2009) discussed differences between aerosol microphysical modules under stratospheric conditions, but their simulations were done using box models with high initial SO2 concentrations (no continuous supply as in case of SAI).

In section 2.1 it now reads: *"However when using M7, this requires changes in the configuration of the mode and having a narrower width of the largest mode to improve representation of the stratospheric aerosols (Kokkola et al., 2009). Thus one downside of using a modal scheme is that tropospheric and stratospheric aerosols are not well described with the same mode configuration."*

We also added the following paragraph to conclusions (includes suggestions from other reviewer) :

*"There are several factors which support a sectional model over a modal model for stratospheric aerosol simulations, despite the fact that the modal scheme is significantly computational faster than sectional (simulations with M7 were 60% faster than SALSA). First of all, tropospheric and stratospheric aerosols require different configurations for modes and thus studying both in the same simulations is not recommended. In addition, even though only stratospheric aerosols are studied, the tropospheric aerosols, which were not well represented by configuration designed for stratospheric aerosols, can affect indirectly to stratospheric aerosols. In SAI simulations, and especially in the case of continuous injections, the size distribution inside the injection region does not have a clear multimodal structure in the sectional model simulations except for the lowest injection rates (1-2 Tg(S)yr-1) (Fig 5 and Fig A3 ) (English et al, 2012, Kleinschmitt et al., 2018). This is probably because there is available H2SO4 gas for particles to grow by condensation, and particles are not accumulating to certain size classes by coagulation. This kind of size distribution cannot be represented by 4 modes and in this study the problem culminates in that there is a persistent gap between the two largest modes. One option could be to use more modes, but then the computational benefits compared to sectional schemes would become smaller. In the standard setup of M7 (the largest) coarse mode width is 2.0 instead of 1.2 which is used here. This would make the gap between the two largest modes smaller. However in the case of stratospheric sulfur injections or a large volcanic eruption, a wider coarse mode width leads to a tail of large particles. This causes an overestimation of the effective radius of the coarse mode and increased the sedimentation velocity and reduced residence time of aerosols in the stratosphere which is the reason why the different setup is used for stratospheric aerosols. One option could be to increase mode widths of the Aitken and accumulation modes. However, number concentrations of these modes are typically higher and thus widening of the modes can lead to a situation, where widened mode would cover the adjacent larger mode. It is also good to keep in mind that the partitioning of sulfuric acid to particle phase due to nucleation over condensation was suspiciously large in SALSA and the model produced significantly larger net total radiative forcing than in e.g. Kleinschmitt et, al. (2018), where simulations were done with the sectional model. Thus, even though there was not as clear a shortcoming as the gap between modes in M7, there is a need to analyze the individual microphysical processes and to understand the differences between the results of different sectional models. "*

**In terms of the analysis related to this issue, I found the crucial aerosol number size distribution plots which show this issue in the main text very busy and difficult to follow. Figures 2 and 5 had many overlapping features and were not at all intuitive,**

**whereas the design choice for figure S5 was much, much clearer. Would it be possible to show light scattered and LW absorbed by aerosols as a function of aerosol size? If not, is there a way to plot the scattering like the absorption? In either case, a much clearer way to show these results and the implications of the modal gap is needed, i.e. something more like figure S5.**

We remade Fig. 2 to be similar to Fig S5. With SALSA bins included the figure is quite busy, but as there is a clear pattern of how the size distribution evolves with increased injection rate, we think that the figure is not too difficult to follow. We decided not to include scattered radiation. It is not straightforward to define scattered radiation as a function of size as it depends on several different factors (composition of aerosols, radiation wavelength and intensity, solar zenit angle etc.) and we think that it does not give significantly more additional information compared to existing "optimal backscattering band". We also wanted to draw the reader's attention to the gap between modes, and why it matters, and for that we think that "optimal backscattering band" illustrates this best.

[Figure]

**Relatedly, the authors devote a page to describing the onegb results as an outlier, indicative of some numerical issue, where it seems to me that the disproportionate efficiency of onegb is due to the fact that no modal gap opens up in this scenario (figure S5). Am I wrong to think that it is the other scenarios that show the shortcoming and not onegb?**

This is a good point and actually compared the analyzed results between SALSA and M7, the onegb scenario is the most consistent between models. You are also right in that the "modal gap" has probably a smaller impact on results of onegb than in other scenarios. However the onegb is an outlier in the sense that results change with the increase of the injection rate. As Figure S6 shows (S4 in previous version of the manuscript) , the lifetime of sulfate increases with the injection rate in the case of onegb simulated with M7 which is not seen in any other simulations and is against our common understanding. In addition, the effective radius of stratospheric aerosols and SW radiative forcing efficiency are suspicious (Fig S6 c and d). And as Fig S7 (S5 earlier) shows, there is also a gap in onegb simulations and the mean radius of the accumulation mode is at the highest limit. Note that the name of the scenario Onegb is changed to Point. (*r2)

We now changed the line *"However, if we exclude the Onegb scenario with 50 Tg(S)-1 injection rate, which showed spurious results, probably due to numerical issues, the most efficient SAI scenario was 76 % larger than in the least efficient scenario based on M7 simulations."*
to:
*"However, simulations of the Point scenario with high injection rates (> 20 Tg(S)-1 ) simulated with M7 showed an increase in the sulfate lifetime with increased injection rate. This differs from all other scenarios and the Point scenario simulated with SALSA. If the Point scenario is not taken into account, in M7 simulations the most efficient SAI scenario exhibited a forcing which was 76 % higher than the least efficient scenario."*

**Another general issue with this paper is that the discussion section is a little thin, with much of the detailed discussion of the results occurring within the results section. As most readers won't get into the depths of the results text, the discussion and conclusion should really bring out all the most important discussion points. Some expansion of the discussion, drawing existing text from the results section, would address this.**
We have now expanded the discussion section based on both reviewers comments. However as the discussion section is quite long already we decided to leave some of the detailed discussion in the results section where the corresponding results are discussed. We strove for the main points of the manuscript to be discussed in Conclusions/Discussion.

**I also had a few other general comments that I think need addressing and a long list of minor suggestions that will follow.**

**A longer discussion of Kokkola et al. 2018, which is mentioned briefly in the methods, seems warranted in the introduction as they apply both aerosol modules to simulate Pinatubo, a highly relevant case.**
In the introduction it now reads: *"These modules have shown stratospheric aerosol loads consistent with the observations of the Mt. Pinatubo 1991 eruption (Niemeier et al., 2009, Toohey et al, 2011, Laakso et al., 2016, Kokkola et al., 2018). Here both modules are used to study how the simulated impacts of geoengineering depend on the injection strategy and magnitude and how these results depend on the aerosol microphysical module used. "*

**A wide range of different terms are used for SAI, I'd suggest sticking with ONE and using consistently throughout all text and figures.**
We modified the text so that it is consistently referred to the same term. Please note that in some cases we refer to SRM when the matter can be seen as relevant for SRM generally and not only for SAI.

**The text is generally well-written and clear but there are definitely some English grammar and phrasing issues that could be picked up with a careful proof-read.**
We tried to correct this issue. The final text will be polished off by the ACP copy-editing team.

**L33 – "changes the structure" – in what sense?**
We modified this line as following: *"Even though SRM could, in theory, be used to mitigate or even compensate the global mean net radiation flux changes due to GHGs, SW and LW*

*radiative fluxes are zonally and vertically still different compared to those in the unperturbed atmosphere"*

**L34-36 – make clear that this is relative to no change, i.e. it is not still warming relative to a case without geoengineering.**
We added:
*"..and can lead to the cooling of the tropics while high latitudes are still warmer than before GHG-induced warming, if solar radiation is reduced uniformly (e.g. percent solar constant reduction)."*

**L42 – which have**
Fixed.

**L47 – is the lack of measurements the leading driver? Are there others?**
**L49 – most rely on models: which don't and are they worth taking seriously?**
We modified these lines as following: *"Because of this nonlinear nature of aerosol evolution, together with a lack of measurements after large volcanic eruptions, the climate model simulations are required to understand climate impacts of SAI."*

**L58 – I'd suggest rephrasing this and the previous claim so they are compatible, i.e. if tilmes simulated aerosol microphysics but found a linear effect then the previous statement is misleading.**
True. Following line is added:
*"However this relation was defined based on scenarios where the background conditions were not fixed and injection strategy was changed during the simulation (see e.g. Visioni et al 2020), and thus these results are not directly comparable with the above mentioned studies."*

**L60 – I'd suggest adopting consistent terminology throughout, there is no need to use SAI, stratospheric geoengineering and stratospheric sulphur injections.**
SAI is now consistently used.

**L71 – "both LW and SW radiative forcing, which have opposite impact on net radiation" this isn't universally true but true of those forcing terms for SAI.**
"Have" changed to "had".

**L91 – larges = larger**
Fixed

**L96 – at a certain**
Fixed

**L109 – this sets limitations**
Fixed

**L140 – suggestion: 'the term "radiative forcing" refers to the instantaneous radiative forcing'**
It is now written as suggested

**L146 – are these slight changes included here?**

We rewrote this as following:

*"However, when using M7 this requires changes in the configuration of the mode and a narrower width of the largest mode to improve representation of the stratospheric aerosols (Kokkola et al., 2009). Thus, one downside of using a modal scheme is that tropospheric and stratospheric aerosols are not well described with the same model configuration."*

**L149-166 – This description of SALSA could be clearer on which aspects described are part of the standard SALSA configuration and which are novel to this paper, and how they differ from this standard configuration.**

We think that line *"For this study, we made one change to the definition of size bins in the the default setup in SALSA Kokkola et al (2018)."* was the one that caused confusion especially when "the change" was brought out at the end of the paragraph. The line is now removed and in the end of the section now reads: "A *change in the lower bound of the first subregion was the only change which was made to the default setup in SALSA (Kokkola et al 2018)* "

**L150 – one change made but what was it? Do the next sentences describe the change or the standard configuration?**

-See the previous reply.

**L159 – "new particle formation scheme" new as in new to this paper or new in some other sense? Is this the change?**

This was a bit misleading. We now rewrote this: "*The scheme for new particle formation is based on the..*"

**L163-165 – changed from what to 3 nm?**

To clarify this, it now reads as follows:

*"This was solved by changing the lower bound the size range of the first subregion (three smallest bins) from 3 nm to 1.02873 nm so that the volume mean diameter of smallest bin was the same as the diameter of the newly formed particles (3nm)"*

**L177 – "Our model setup does not include all modifications done in (Niemeier and Timmreck, 2015) e.g. the simple stratospheric sulfur scheme." Have the authors described all the changes, and Niemeier and Timmreck did more or are there additional, unspecified changes that take us some of the way towards what Niemeier and Timmreck did? Not clear that the "simple stratospheric sulphur scheme" refers to, is this described by Niemeier and Timmreck?**

This is now said explicitly: *"Our model setup does not include additional stratospheric chemistry, limitation of available OH for oxidation of $SO_2$ in extreme high $SO_2$ concentration (> 1000 Tg (S)) and forced evaporation of sulfate over 30 km as in Niemeier and Timmreck, 2015 and Niemeier et al (2021)."*

**L184 – does "band" here mean injection occurs across all longitudes, either way make this clear and explain it in table 1.**

This is now explained in the text: *"In our Baseline scenario, sulfur was injected at 20-22 km altitude and a band across all longitudes between the latitudes 10N and 10 S."*

In the table caption it now reads: *"Injections are done across all longitudes between stated latitudes in all other scenarios other than the Point scenario."*

**L187 – narrow, 1.9N – 1.9S – is this +1 / -1 gridbox from equator, if so make that clear and make clear the similarity to onegb.**
Yes and now the text it reads: *"Scenarios Narrow and Wide were simulated to study the impacts of shrinking or widening the injection area. In the Narrow scenario, injections are done between the latitudes 1.9◦N and 1.9◦S (two grid boxes wide band) and to 21 km altitude (one model vertical level) and in Wide scenario, sulfur is injected between the latitudes 30◦N and 30◦S (20 - 22 km)"*

We also included information about the injection altitude when discussing scenarios Low and High, so that the description in the text is consistent between the scenarios:
*"Scenarios Low (injections at 18-20 km altitude) and High (injections at 22-24 km altitude) were done to study the dependence of radiative forcing on the altitude."*

**L192 – Perhaps clarify, does pulse2m inject 6 months out of the year, i.e. every other months**
To clarify this, we changed the names of the scenarios, extended the description of the scenario and changed the order of the sentences in the paragraph which now reads: *"We also simulated two scenarios where injections are concentrated on certain times of the year instead of having continuous emissions over a year. In both of these scenarios the length of the one injection period is one month. In Pulse-eom, sulfur was injected every other month starting from January (6 injection periods per year). In Pulse-Jan-Jul scenario, sulfur was injected during two single months per year, January and July. In these cases, the concentration of sulfur during injections is higher compared to the Baseline scenario which has a constant injection rate throughout the year. Instead, in scenarios Pulse-eom, and Pulse-Jan-Jul, injections are interrupted outside the injection periods. This might affect the size distribution of the stratospheric aerosols"*

In Table 1:
Pulse-eom - 10◦N - 10◦S and 20-22 km, injection in every other month
Pulse-Jan-Jul - 10◦N - 10◦S and 20-22 km, injection only in January and July

**L192-194 – Are pulse2m / pulse6m the most intuitive name for these scenarios? Pulse alt months, pulse twice yearly?**
We agree that "pulse2m" and "pulse6m" were not the most intuitive names and these are now changed to "Pulse-eom" (every other month) and "Pulse-Jan-July". (In addition we changed the name of the "onegb" scenario to "Point")

**L195 – seasonal – specify how it varies through the year, then explain that the authors followed laakso et al. 2017's approach. Readers don't want to have to look up the methods of another paper to capture a detail that can be explained in a few words.**
This is now rewritten as:
*"In Seasonal scenario, 20◦wide injection area is varied gradually between 40◦N and 40◦S throughout the year. The northernmost position (40◦N - 20◦N) of the injection area is in May and the southernmost position (20◦S - 40◦S) is in November (see Fig S1)."*

We also included following figure in the supplement, which shows injection areas in studied simulations:

[Figure]

**L197 – why not call it two grid box or something else? Seems odd to inject into 2 grid boxes and then call the scenario 1 grid box.**

We ended up this name as there injections were done longitudinally to one grid box and "onegb" gives you right impression about the scenario (it doesn't really matter if the injections are done in one or two adjacent grid boxes). E.g. "two-gridbox" might give the impression that injections are done in two different locations. However, we found out after the submission of the manuscript that injections were done into 1 grid box instead of two adjacent grid boxes on both sides of the Equator. Corresponding parts of the text are now fixed.

Nevertheless, this scenario name is now changed to "Point" so that it does not refer to something that is a model characteristic.

**Table 1 – Might be good to specify here or in text the vertical distribution in terms of gridboxes. I presume 21km is the centre of a single gridbox, which has a bottom and top that spans some altitude, and 20-22 km captures 3 or so gridboxes.**

In section 2.2 it now reads: *"In our Baseline scenario, sulfur was injected at 20-22 km altitude (3-4 model vertical levels).."* and *"In the Narrow scenario, injections are done between the latitudes 1.9°N and 1.9°S(two grid boxes wide band) and to 21 km altitude (one model vertical level) ".*

**Figure 1 – This is unclear, do the faint results correspond to the right-hand faint axis. This needs to be made clear in the figure caption!**

This is correct and this was missing in the figure. We now have added: *"Faint colors (forcing efficiency in a)-c) and effective radius in d) ) correspond to the right-hand faint axis."* to the figure caption.

**L221 – in other hand the = on the other hand?**

Fixed.

**L250 – does this (sulfur burden) indicate that (size, number) alone?**

Not alone but with forcing it does. We modified this line as following:

*"Lower SW radiative forcing, higher LW radiative forcing and a slightly shorter lifetime is caused by less and larger sulfate particles in M7 simulations than in SALSA."*

**L284 – 1) Does that imply: shorter residence times meaning fewer particles make it to high latitudes? If so spell that out.**

Yes and this is now written in the text: *"1) when the amount of sulfur was increased, aerosols became relatively larger, thus having higher gravitation settling velocity which means that fewer particles made it to high latitudes"*

**L285 – This sentence is a bit of a mess, how about: "as it has been shown (visioni…)," sulfate aerosols à tropical warming à strengthened polar vortex à reducing (not preventing) transport of particles to polar stratosphere**

This is now rewritten: *"as it has been shown, injected sulfate aerosols cause tropical stratospheric heating and a strengthening of the polar vortex, which reduces aerosol transportation to the polar stratosphere"*

**Figure 3 – position of a / b not consistent. Choice of colorscale a bit odd, going from green to cyan to green again. Why not have a simple perceptually uniform, colorblind-friendly sequential colormap like viridis (https://matplotlib.org/stable/tutorials/colors/colormaps.html). Y-axis label: normalized SW forcing, would be snappier and could be explained in caption.**

Positions of a and b are now consistent. As we want to keep colors indicating SALSA (blueish) and M7 (reddish) results separate throughout the entire manuscript, we chose a diverging colormap (*spectral*) and used one half of the colormap for SALSA and the other half for M7. This colormap is now also changed in Figures 4, 9 and A2, revised Figures 2 and 5, supplement Figures S2 and S7 (S1 and S5 in the review version).

**Figure 2 and 5, legend to top-right? Presumably the magenta line is unitless but linear rather than logarithmic like the Y-axis, but this should be specified.**

The legend in Figure 2 is now put on the right side of the figure. It is not a conventional legend anymore, but we think that this one works better here. In Figure 5, legends are now in the middle on the top of the plot. Captions of the figures now read: *"Magenta line shows (unitless) the relative dependence (in a linear, not logarithmic scale) of absorption of 8000nm wavelength on the (dry) diameter of the sulfate aerosols, and is based on the radiation calculation module of SALSA."*

**Figure 5 – Is this the best way to show this? I find Figure 5 very difficult to follow as there are so many elements and I'm comparing left-right panels. Perhaps reconfigure so that the relevant difference (results at one latitude vs the other) can be seen legibly in individual panels (more than 2? Or for only 5 or 50 not both?)**

Now there are still two panels, but 5 Tg(S)/yr case is in Fig 5a and the 50 Tg(S)/yr case in Fig 5b. So now the one panel shows one latitude vs the other.

[Figure]

[Figure]

**L306-307 – Is the deposition process slower, implying for the same mass it falls out more slowly, or was there less to deposit so the total rate is lower?**

Good point. "Slower" is now changed to say that the deposition rates were lower. They are probably lower because there is less aerosol mass to deposit (sulfate will sediment anyway at some point), but we don't want to explicitly speculate it here as these processes are difficult to quantify.

**L308-311 – perhaps rephrase to remove some of these caveats and details, these sentences are a little tangled.**

These lines are removed.

**L324 – can this 10nm particle effect be explained briefly here?**

This is now rewritten as follows: *"In addition, in SALSA there is a condensation sink due to a high number of particles smaller than 10 nm which does not exist in M7 and thus there is less gaseous sulfuric acid to condense to the larger particles."*

**L334-334 – not clear what is meant by nucleation over-running (out-competing?) condensation, or how figure 4 supports that view.**

These lines are now rewritten: *"when nucleation takes place, it out-competes condensation and there is significantly less condensation. This conclusion is supported by Fig. 4a, which shows that there the effective radius is clearly smaller inside latitudes where injections take place, compared to higher latitudes"*

**L339-345 – Describing these supplementary results quantitatively would help to make clear the significance of this effect relative to other differences.**

This is a good suggestion. These lines are now rewritten as follows: *"With the original setup, 5 Tg(S)yr-1 or higher injection rates simulated with SALSA showed 137-147% larger global mean all sky net radiative forcing than M7. When nucleation was replaced by injecting 25% of injected sulfur as 3nm particles, radiative forcing was now only 78-99% larger in SALSA than M7. Thus, these simulations showed that excluding nucleation brought the global mean net radiative forcing results between the models closer to each other, although a significant difference remained. This is because there is not a large difference in aerosol number size distribution for..."*

**L346-L364 – Proof-reading for English grammar and phrasing is needed here.**

We tried to make these lines more clear for the reader.

**L369 – SRM? Stick to one term**
Changed to "SAI"

**L361-364 – Should this be stated more strongly? The modal representation seems to have a substantial shortcoming, a gap just where the aerosols are most effectively scattering, which means that it surely under-estimates the radiative forcing from SAI.**
Based on these results, it is very likely expected that the M7 simulated radiative forcing is underestimated because of the representation which leads to a gap in the size distribution. We can also say that the gap is the one of the main factors behind the difference between SALSA and M7 results. But we cannot say for sure that M7 results are underestimated because of the gap, because we don't know what we could get with a "better" setup and because we don't have the observations to compare to. Note also that radiative forcing from the simulations of sectional model in Kleinschmitt et al, were even lower than in our modal model simulations.

**L365-368 – Seems to change topic here, why not open in a way that makes clear where this new paragraph is heading, i.e. simulations of volcanic eruptions do not expose this shortcoming of M7.**
We added *"The above-mentioned differences in the responses between models can easily go unnoticed when models are evaluated against measurements after a large volcanic eruption."*
at the beginning of the paragraph and
*"The gap in the size distribution is also widened as a consequence of continuous injections band across all longitudes when the accumulation mode cannot grow while the coarse mode is getting larger due to continuous injections. The gap is narrower in case of point-like injections as we see in the next section. This also indicates that such a clear gap as in our Baseline scenario does not occur in a simulation of the large volcanic eruption."*
to the end of the paragraph.

**L383 – This clear-sky comment could perhaps be better made in the methods, as the clear-sky analysis is the focus across much of the paper rather than a novelty of this section.**
We decide to keep it here as all-sky radiative forcings are shown in Fig 1.

**Figures 6 and 7 – I wonder if there's a simpler / clearer way to plot this. Perhaps, rather than scattering the scenarios as coloured lines along an injection amount X-axis, flip this with the scenarios laid out along the X-axis (like a bar chart?) and the injection amount as coloured points scattered up the y-dimension. Just an idea.**
That is definitely one nice way to do it. However, here with these figures we wanted to show if the mutual relation of mean radiative forcing, burden and effective radii of scenarios changes with the injection rate and for that, we think the existing Figures are more appropriate.

**Figure 8 – the seasonal overlay obscures more than it reveals, I'd suggest relegating it to a supplementary figure and referencing it in the methods (where it is unclear how the seasonal strategy is implemented). Having a conventional legend would be more**

**legible, and the blue arrows are unnecessary. There's a lot of lines here, narrowing the y-range would make them more distinct**

The y-range was defined based on the zonal mean clear sky radiative forcing which was meant to show in the figure, but the actual data was accidentally the all-sky radiative forcing. This is now fixed and the figure shows now clear sky radiative forcing. Lines are now more distinct. A figure with all-sky data is added to supplement (Fig S5). The blue arrows are removed. The legend is changed to a conventional one. However we wanted to keep all (including seasonal) indicators (shaded areas) in the figure so that it would be easier to see the relationship between the injection area and the resulting radiative forcing.

Legend in the figure 9 is also changed to conventional.

[Figure]

[Figure]

**L410-412 – so the heating effect is more concentrated?**
We removed the mention about the role of heating and increased uplifting as the change in the residual vertical velocity in Figure S12 (revised supplement figures) does not unequivocally show that.

**L423 – "less aerosols over the equator are under continuous injection" – isn't clear what this means.**
This is now rewritten to be more clear:
*"However, as injection takes place longitudinally in an area which is one grid box wide, existing aerosols over the Equator from previous injections are not, most of the time, condensation and coagulation sink for injected sulfur in the injection area, as is the case when injections occur across all longitudes."*

**L427 – Best to explain how they are different or simply remove this sentence and give the results.**

This was explained in the following sentence. We changed the dot between these two sentences to colon:

*"Results of the Point scenario with M7 showed significantly different behaviour when increasing the injection rate compared to the other scenarios and even to the same scenario when SALSA was used instead of M7: While the clear sky global mean net radiative forcing was roughly.."*

**L437 – This last sentence isn't clear.**
This is rewritten:
*"In addition, SW forcing efficiency did not decrease and the effective radius of stratospheric aerosols did not increase gradually with the injection rate as in simulations with SALSA and in all the other scenarios."*

**L442 – accumulation mode, grow.**
Fixed

**L445 – would be clearer to state what Niemeier and Timmreck found.**
This is now stated: *"This also contradicts the results of Niemeier and Timmreck (2015) where the size of the coarse mode increased with the injection rate in a simulation of injections to one grid box."*

**3.3.2 – it seems like the issue is more with the normal scenarios than with onegb as a gap opens up between the accumulation and coarse modes in the normal scenarios and not in onegb.**
As we mentioned in our earlier reply, the gap is present also in onegb scenario, but as you said, it is not as distinct as in other scenarios.

**L468 – This isn't informative unless the difference in the untuned version is made clear, and its significance explained.**
This line is removed.

**L532-533 – this nice, clear description of the pulse scenarios should appear in the methods section.**
We have now made the description (and names) of pulsed scenarios more clear in the methods section.

**L540 5% larger?**
We added *"longer"*

**L543 – should that be 6 times greater – 12 months of emissions in 2 months?**
Yes, fixed.

**L545-546 – rephrase this sentence.**
This is rewritten as follows: *"Thus, when a new injection period started, there were less particles from the preceding injections present over the Equator. Thus, condensation on existing particles and coagulation between the new and old particles are lower than in case of continuous injections."*

**L560 – explain the strategy in more depth, then cite laakso.**
Now it reads:
*"Last of the studied scenarios was one where a 20 wide injection area is varied gradually between 40 N and 40 S during the year. As suggested in Laakso et al 2017, the aim of seasonally changing strategy is to increase the efficiency of SAI compared to continuous Equatorial.."*

**L568 – compared to baseline or is it a different equatorial scenario?**
*"Equatorial injection"* changed to *"baseline scenario"*

**L572 – distributed evenly would be a fairer description, concentrated in mid-latitudes suggests the forcing is much greater there than elsewhere.**
This is true. However, as mentioned earlier, the all sky radiative forcing instead of clear sky was accidentally plotted in Fig 8 and what was said in L572 is true for clear sky forcing.

**L573-585 – This paragraph needs some rewriting, I'd suggest opening in a way that makes it clear what the paragraph is driving at.**
We tried to make it more clear.

**L580 – the previous paragraph reports increases of at least 30% which is inconsistent, unless I misunderstood.**
Here we compare our 5 Tg(S)/yr injection rate seasonal simulations to corresponding simulations in Laakso et al (2017). Here our simulations show 10% increase compared to the baseline scenario.  We clarify this: *"However in this study the simulations of seasonal injection with 5 Tg(S)yr−1 and with SALSA led to a 10 % stronger radiative forcing compared to Baseline scenario."*

**Figure 11 – plotting control last would make it easier to see some of the changes.**
CTRL is now on the top.

**L599-611 – might be worth extending the plot to include more (all?) of the troposphere and mentioning the substantial decrease in vertical velocity in the upper troposphere seen in the simulations with large warmings, looks like it's down by a third in the 100TG M7 case.**
The y-axes is extended to 12 km altitude. We added the following text in the manuscript:
*"While residual vertical velocity increases above tropopause, in the upper troposphere the residual vertical velocity decreases with the simulated injection rate. This reduction is larger in M7 than SALSA  and simulation of 100 Tg(S)yr−1injection rate with M7 showed 35% lower residual vertical velocity at 15 km altitude compared to CTRL simulation"*

**L620-622 – is there a consistent temperature threshold between the 2 cases?**
We added: *"This corresponds to stratospheric temperature anomalies compared to the CTRL scenario, which were the same magnitude in those two simulations."*

**L666-667 – "2) limiting the evolution and the shape of the particle size distribution due to mode setup in M7." This isn't clear, perhaps it should be expanded a little here.**

This is rewritten to be more clear to read: *"a local minimum in aerosol number size distribution between two largest modes, caused by repartitioning of particles between the modes in model, which coincidences with the optimal particle size for backscattering"*

**L669-671 – The phrasing here suggests this effect is large, could it be quantified? What fraction nucleates vs. coagulates in these 2 models?**

Unfortunately nucleation and condensation rates were not model outputs and thus we cannot quantify these. We changed this part a bit as it earlier stated that nucleation rate is high in SALSA, but even though sulfuric acid tends to form new particles rather than condense on the existing one, it does not mean that it is caused by a too high nucleation rate: *"In the stratosphere, new particle formation by nucleation is fed by continuous injections. On the other hand, there are already pre-existing particles from the preceding injections to which injected sulfur can condense. Thus, there is a competition between these two processes for available sulfuric acid gas. In SALSA, sulfuric acid tends to form new particles rather than condensing on the pre-existing ones while in M7 simulations the opposite is true. "*

**L673-675 – This "modal gap" is critical and should be elaborated in a full paragraph. One of the take-aways for the community from this paper ought to be to watch out for this phenomena in their own modal model setups. Some more discussion of whether this "feature" of the modal setup is realistic, presumably not, and what it implies about the applicability of these models in future is needed. Does this imply that the modal model is significantly underestimating the effectiveness of SAI? Does this imply that modal models need a minimum of 5 modes to represent SAI? Are there some other design choices that could be made to avoid this issue or is it a fundamental weakness of modal models, and an argument for the application of sectional models?**

This is a good suggestion and we added one new paragraph about this (see our second reply). We don't want to make direct suggestions to use either one of these models, or modal vs. sectional models generally. The model capability to simulate stratospheric aerosols depends on several different factors e.g individual microphysical processes in addition to how aerosol size distribution is represented and those should be analyzed comprehensively. We don't want to estimate if the model underestimates radiative forcing either, even though our results indicate that, because there is no observational data to compare with. Nevertheless, we hope that shortcomings found here are taken into account by the community and that more attention is given to aerosol microphysical modelling in SRM modeling studies.

**L690 – should make clear that this is sufficient to return forcing to pre-industrial levels, as readers may assume that the authors mean change relative to today.**

This is now fixed and we also change the term "radiative imbalance" to "radiative forcing" in this line: *"compensating all radiative forcing in the Representative Concentration Pathway (RCP) 2.6 from preindustrial period to the end of this century (2.6 Wm-2) by stratospheric sulfur injections"*

**L688-699 – Is it fair to describe these differences as uncertainties when the authors have revealed a serious shortcoming in one of the models?**

This is true. However we also wanted to discuss also a little bit more on a general level what microphysical uncertainties mean. We changes first line of paragraph to: *"Shortcomings and uncertainties in microphysics will further lead to large uncertainties in estimating"* and line *"These uncertainties between models can have also a significant impact on uncertainties related..."* to *"These differences between model results can have also a significant impact on uncertainties related..."*

**L704-705 – Mention which were the most and least efficient scenarios, and perhaps briefly why.**
We added the end of this paragraph: *"Based on results from both models, three most efficient scenarios were Point, Pulse-Jan-Jul and Seasonal. Common to all of these scenarios was that instead of a stable injections to band across all longitudes, injections were done either by injecting only to one model grid box, suspending injection for 5 months before the next injection period, or changing injection area seasonally."*

**L707-709 – As I suggested before, it seems like Onegb was only spurious in that it didn't cause M7 to exhibit the modal gap issue. I'd argue that this suggests M7 is only capable of reasonably simulating small deployments, before modal gap opens up, and onegb where it doesn't open up.**
See our third reply (*r2). We changed these lines so that instead of calling results of Onegb "spurious" we explain how it differs from all other simulations.

**L710-720 – It seems to me that a reasonable take-away from this paper is that equatorial deployments are inefficient and result in an over-concentration of aerosols in the tropics, and that wide, seasonally-shifting or extra-tropical (other papers) deployments produce more efficient, more even forcing and hence are likely preferable. The implication then would be that future SAI studies should consider abandoning the equatorial strategy (which was chosen for simplicity, and features in G6sulfur) in favour of one of these alternatives (in a future round of GeoMIP?).**
We added one paragraph on this to Conclusions and Discussions. See our first reply (*r1)

**L725-728 – Isn't the key the relative concentration of large particles vs. gaseous SO2? Onegb and narrow both deploy into an existing plume but differ in that the concentration of gaseous SO2 is ~100 times higher in onegb promoting nucleation.**
This is also true. However, we think the key factor why Onegb was much more efficient than Narrow was that the situation for a formed new particle is different when it travels across longitudes at the Equator. It experiences (or be very close to) continuous injections and thus is a condensation sink for fresh H2SO4 gas from the injections in Narrow while in Onegb particle travel farther from the injection area where there are less gas. Note that injected sulfur concentration was higher in Narrow than Pulse6m (Pulse-Jan-Jul in revised version), but still Narrow scenario exhibited a significantly lower forcing. In Pulse6m new particles experience injections not more than one month until injections are suspended. When they start again, most of the aerosols are moved out from the injection area.

**L729 – explain the baseline distribution, with its disproportionate forcing in the tropics, before the differences.**

We added to the text: *"Equatorial injections in our Baseline scenario resulted in maximum zonal forcing over the tropics. This relative disproportion of radiative forcing between low and high latitudes was increased with the higher injection rate. "*

**L735-736 – There is no warming at high latitudes due to SAI, it offsets ~80-90% of the warming from GHGs rather than 100%**

We rewrote these lines as: *"Several studies have shown that offsetting the mean GHG-warming with uniform SRM or Equatorial injections can lead to overcooling of the tropics and warming at high latitudes. This could be prevented by seasonally varying sulfur injections and without any trade off in the total radiative forcing, as.."*

**L739-741 – This is the only mention of the dynamical changes in the discussion. The authors gave a whole sub-section to the topic, seems odd to not give it at least a paragraph in the discussion and conclusions.**

We disconnected these lines to their own paragraph and added a couple of new lines about dynamical changes: *"We also studied dynamical changes in the stratosphere. As M7 produced larger aerosols and higher absorption of LW radiation, warming in the stratosphere was stronger in M7 than SALSA simulations. Thus the increase in residual vertical velocity was larger and the slowing down of the QBO was more significant in M7 than in simulations with the corresponding injection rate with SALSA. In our Baseline scenario the QBO was vanishing with the injection larger than 10 Tg(S)yr−1 injection rate based on simulation with M7 while in the simulation with SALSA, more than 20 Tg(S)yr−1 was required to shut down QBO. Based on SALSA simulations, the Seasonal scenario did not have a significant impact on the slowing down of the QBO even with 50 Tg(S)−1 injections."*

Other corrections:
1) "and forcing efficiency" is removed from Figure A1 panel titles
2) Caption of Fig A3 is fixed, colors are changed and number distribution of M7 simulations is now shown as sum of the modes (thus not showing individual modes separately).
3) We added a comment to Section 2.2 that concentration of injected sulfur varies in the seasonal scenario during the year: *"Note that as injection band is always 20° latitudes wide and a same mass in injected in every month, the concentration of injected sulfur is smaller during the times when the injection area is located over the Equator compared to times, when it is over midlatitudes"*